# Three Phages One Host: Isolation and Characterization of *Pantoea agglomerans* Phages from a Grasshopper Specimen

**DOI:** 10.3390/ijms24031820

**Published:** 2023-01-17

**Authors:** Nikita Zrelovs, Juris Jansons, Tatjana Kazaka, Andris Kazaks, Andris Dislers

**Affiliations:** Latvian Biomedical Research and Study Centre, Ratsupites 1 k-1, LV-1067 Riga, Latvia

**Keywords:** *Pantoea agglomerans*, phages, insect-associated phages, podophages, myophage, whole-genome sequencing, *Autographiviridae*, *Lietduovirus*, *Eracentumvirus*

## Abstract

The bacterial genus *Pantoea* comprises species found in a variety of different environmental sources. *Pantoea* spp. are often recovered from plant material and are capable of both benefitting the plants and acting like phytopathogens. Some species of *Pantoea* (including *P. agglomerans*) are considered opportunistic human pathogens capable of causing various infections in immunocompromised subjects. In this study, a strain of *P. agglomerans* (identified by 16S rRNA gene sequencing) was isolated from a dead specimen of an unidentified Latvian grasshopper species. The retrieved strain of *P. agglomerans* was then used as a host for the potential retrieval of phages from the same source material. After rounds of plaque purification and propagation, three high-titer lysates corresponding to putatively distinct phages were acquired. Transmission electron microscopy revealed that one of the phages was a myophage with an unusual morphology, while the two others were typical podophages. Whole-genome sequencing (WGS) was performed for each of these isolated phages. Genome de novo assembly and subsequent functional annotation confirmed that three different strictly lytic phages were isolated. Elaborate genomic characterization of the acquired phages was performed to elucidate their place within the so-far-uncovered phage diversity.

## 1. Introduction

The bacterial genus *Pantoea* includes multiple Gram-negative, yellow-pigmented rods that are frequently isolated from a variety of environmental and higher-organism-associated sources [1], hence the genus name derivation from pantoîos (Gr. masc. adj.)—“of all sorts and sources”—with a meaning that these bacteria are retrievable from diverse geographical and ecological sources.

*Pantoea* spp. are still, arguably, mainly thought of as bacteria associated with plants, capable of both having positive effects on plants (e.g., [2,3,4,5]) and acting like phytopathogens causing diverse diseases in various plants (e.g., [6,7,8,9]). These bacteria are also considered to be opportunistic human pathogens associated with various conditions in immunocompromised individuals (e.g., [10,11,12,13,14,15,16]).

*Pantoea agglomerans* is the type species of the genus represented by a strain ATCC 27155 isolated from a knee laceration [17]. Throughout the years, a body of evidence documenting multiple effects, both beneficial [18] and deleterious [14], has been presented for isolated strains of *P. agglomerans*. Multiple potential applications of *P. agglomerans* strains are being explored due to their environmental versatility in conjunction with their ability to promote plant growth, function as a biocontrol agent by antagonizing with fungal and bacterial pathogens, and show biodegradative capabilities [1]. The potential pathogenicity of the *P. agglomerans* strains, however, requires their in-depth characterization to evaluate the suitability of a given strain for particular practical applications.

Viruses of bacteria—bacteriophages (phages)—are omnipresent in every environment where bacteria thrive. As natural predators of bacteria that have the ability to rapidly propagate themselves upon invading a susceptible host bacteria population, phages can be harnessed as biocontrol agents to eliminate unwanted bacteria. However, phages generally have a very limited host range, which might be as narrow as several strains of a particular bacterial species. Thus, isolation and characterization of new bacteriophages capable of infecting representatives of the bacterial genera or species even of modest economic or healthcare importance are of particular interest, especially in a case such as when a host has none or just a few bacteriophages yet known to infect it. In addition to the expansion of the knowledge on the uncovered phage diversity, which is of fundamental importance, such studies might potentially provide a good lead for further practical application evaluation of a given novel phage or its proteins.

Besides the three phages described in this study, currently, there are 19 *Pantoea*-infecting phages that have their complete genomes publicly available. These phages represent all the tailed phage morphotypes (myo-, sipho-, and podo-), and their genomes range from 36,790 to 149,913 base pairs, showing a percentage GC content of 39–55.35%. Based on the complete genome-associated metadata, strains of *Pantoea agglomerans* are indicated as the specific host of these phages most commonly (n = 12) followed by *P. deleyi* (n = 3) and *P. dispersa* (n = 2). Eleven of the so far sequenced *Pantoea* phages have been isolated from plant sources (specifically four of the phages that originated from *Ribes* Jostabeere, and another four from *Amelanchier spicata* material). Additionally, five *Pantoea* phages were previously retrieved from water-associated sources and another three from soil samples. This makes the phages Nifs112, Nufs112, and Nafs113 described in this study the first three *Pantoea* phages to be isolated directly from an insect-associated source (Table 1).

## 2. Results and Discussion

### 2.1. Identification of the Host

When streaked on LB agar (1.5%), isolate LS5-2, the isolation host of phages Nifs112, Nufs112, and Nafs113, formed beige colonies at all of the three incubation temperatures tested (RT, +30 °C, and +37 °C). The fastest growth was observed at +30 °C, and the slowest was observed at RT. After their appearance, the LS5-2 colonies were initially beige but turned yellow-orange during the subsequent incubation.

Based on the nearly complete 16S rRNA gene sequence phylogeny reconstruction, strain LS5-2 was identified to belong to the bacterial genus *Pantoea*. More specifically, as LS5-2 shared a well-supported most recent common ancestor with *P. agglomerans* DSM 3493 in the phylogeny, and both sequences had only a few differences, the bacterial isolate LS5-2 was considered to be a strain of *P. agglomerans* (Figure 1).

### 2.2. Plaque and Virion Morphology of the Isolated Phages

All three of the studied phages demonstrated different plaque appearances on the lawn of their isolation host *P. agglomerans* LS5-2 after incubating the double agar overlay plates for ~24 h at room temperature, although each phage showed a considerable possible plaque-size variation (Figure 2, top row).

The plaques formed by Nifs112 were very turbid in appearance and were up to 2 mm in diameter. After incubating the double agar overlay plates with Nifs112 for a longer period of time, it started to seem that the negative colonies themselves were extremely small, and the assumed “plaque” of 2 mm diameter might have been mostly a halo.

The plaques formed by Nufs112, on the other hand, were clear in appearance, but closer inspection revealed a slightly opaquer halo that was ~3.5 mm in diameter from the plaque center, with the plaques themselves having a diameter of around 1 mm.

Nafs113, however, seemed to form rather clear negative colonies of up to 2 mm in diameter without any distinguishable halo, but the majority of the Nafs113 plaques were ~1 mm in diameter.

Transmission electron microscopy revealed that both Nifs112 and Nufs112 were podophages that seemed to differ mostly in their short non-contractile tail appearances. Measurements of six randomly selected virions of both phages revealed that the Nifs112 capsid diameter might have been slightly larger (57.2 ± 2.4 nm) than that of Nufs112 (54.4 ± 1.8 nm). With the opposite holding true for tail lengths, the tail of Nifs112 was measured to be around 11.2 ± 1.6 nm in length, whereas Nufs112 had a tail of 12.8 ± 2 nm. However, a more detailed TEM examination involving at least several tens of virions, additionally taking the planes at which the virions were seen into account, would be necessary when drawing conclusions on whether these capsid size and tail length differences were meaningful. What stood out in the micrographs, however, were the tail appearances, with Nifs112 having seemingly slimmer tails in contrast to the “chunky stubs” of Nufs112 (Figure 2, bottom row).

Halo formation around a plaque usually is explained by depolymerase activity that is associated with the phage-encoded tail spike protein (or proteins) that has been observed on numerous occasions for a plethora of different podophages [27]. Combined virion morphology and plaque appearance observations allowed us to expect the likely presence of depolymerase domains in the tail proteins of both Nifs112 and Nufs112. The activity of such depolymerases on the cells of *P. agglomerans* LS5-2, however, is expected to be highly different based on the differing Nifs112 and Nufs112 negative colony appearances under the same experimental conditions.

Although having an unremarkable plaque appearance, Nafs113 virions were determined to have a less-common myovirus morphotype—an elongated head with an approximately 3:1 length (121.4 ± 6.8 nm)-to-width (43.4 ± 3.8 nm) ratio, to which a 88.5 ± 3.2 nm long contractile tail with a width of 15.4 ± 0.6 nm (uncontracted state) was attached (Figure 2, bottom row).

### 2.3. Complete Genomes of the Isolated Phages

The complete genomes of the isolated *Pantoea* phages were successfully sequenced and de novo assembled with mean depths greater than 200×, and each of the bases covered at least 106, 324, and 30 times for the Nifs112 (46,202 bp), Nufs112 (45,951 bp), and Nafs113 (75,899 bp) complete genomes, respectively (Table 2).

The exact genome physical molecule termini that were predicted for Nifs112 and Nufs112 could be successfully verified by Sanger-based sequencing with custom primers designed to allow reaching the leftward and rightward terminus of the respective genome. No read pile-up signal representative of any of the exact phage genome termini types, however, was detected from the Nafs113 NGS library reads. Thus, it seems that the concatemeric phage DNA likely serves as a packaging substrate for the *Pantoea* phage Nafs113, which might use a headful genome packaging strategy without an identifiable preferred packaging series initiation site/sites, resulting in each virion having a terminally redundant circularly permuted genome.

Genome functional annotations revealed that slightly less than half of the total 59 ORF products predicted in the genome of the Nifs112 encoded hypothetical proteins, for which no functional assignment could be made with confidence (Appendix A). Similarly, 35 out of the predicted 67 proteins putatively encoded by Nufs112 ORFs remained “hypothetical proteins” (Appendix A), whereas for Nafs113, 70% of the predicted 130 ORFs coded for hypothetical proteins (Appendix A) without any hints at their functions, despite the multiple annotation approaches used (conserved domain search, BLASTP, and a more sensitive HHpred approach that is often able to help with protein functional assignments even in the absence of highly homologous sequences). The absence of lysogeny-associated genes within the genomes of either of the three phages allowed us to classify them as being strictly lytic, which was consistent with the microbiological observations during the handling of these three phages.

During the genome annotation step, both the podophages (Nifs112 and Nufs112) were found to belong to the family *Autographiviridae* due to the presence of a single ORF-encoding DNA-dependent RNA polymerase in their genomes (unlike the *Schitoviridae* podophages, harboring three DNA-dependent RNA polymerase genes, one of which is coding for a large virion-associated RNA polymerase [28]). However, despite being isolated on the same host and sharing morphological similarities, a BLASTN comparison of the Nifs112 and Nufs112 complete genomes gave a query coverage of only 7%, indicating a very distant evolutionary relationship between them.

Roary (v. 3.13.0; [29]) pangenome analysis was performed at different BLASTP protein similarity thresholds (95%, 50%, 30%) to identify whether there might be some conserved proteins of interest encoded by the studied phages, which, as was evident at this point, only had a host in common, although a sufficient amount of protein homologs were expected to be present in both *Autographiviridae* podophages. No protein, however, was found to be present in more than a single studied phage at the 95% identity threshold. Lowering the threshold to 50% only identified five proteins that were shared between Nifs112 and Nufs112—a terminase large subunit, phosphatase, a HTH domain-containing protein, and two hypothetical proteins. Setting an identity threshold to 30% revealed that phosphatase was the only protein that could be considered to be shared by all three of the phages. As expected, the lowest threshold used (30%) also revealed that Nifs112 and Nufs112 might have up to 20 homologous proteins, i.e., the ones involved in either virion morphogenesis (major capsid, portal, scaffolding, tail tubular, and tail fiber proteins) or those responsible for nucleic acid metabolism, modification, and repair (e.g., ligase, DNA primase/helicase, DNA polymerase, exonuclease, and recombination endonuclease), and several other, mostly hypothetical, proteins. At this threshold, a homolog of HNH endonuclease from Nafs113 (ORF115 product) was also identified in the Nifs112 proteome (ORF30 product).

After functional annotation of the studied phage genomes, termini types were also additionally verified using the TerL phylogeny reconstruction approach based on a dataset of Merrill and colleagues [30]. The TerL sequences of studied phages reliably fell within the clades comprising TerL sequences from the selected phages that had their packaging strategies/genome termini types verified experimentally, despite them representing only a fraction of the TerL/terminase sequence diversity [31] (Appendix A). After previous experimental verification, it was not much of a surprise that the TerL sequences of Nifs112 and Nufs112 had clustered together with T7-type short direct terminal repeats (SDTR)-employing phages in this packaging strategy prediction tree.

Yet, the TerL phylogeny reconstruction showed that the TerL of Nafs113 clustered together with Sf6-type headful-packaging-employing phages. This further supported our assumptions regarding the Nafs113 headful genome packaging, as, for example, the *Hamiltonella* phage APSE-1 representing the same TerL sequence clade also has a terminally redundant and circularly permuted genome, with the sequence and electron microscopic analysis carried out indicating that “most likely the permutations are randomly distributed over the genome” [32]. Researchers studying the *Shigella* phage Sf6 noted “little evidence for specific termini”, but they also found out that Sf6 DNA appeared to be cleaved at many sites within a large region of about 1800 bp, including a possible *pac* site for the initiation of a packaging series from a DNA concatemer [33]. The extent of the possible amount of the *Pantoea* phage Nafs113’s genome’s circular permutation (e.g., is the packaging series initiated truly randomly or is there still a preferred region, etc.), however, was not further looked at experimentally at the time being.

Although all three of the phages were retrieved using the same host bacteria strain, their genome GC% contents were quite different, with only the Nafs113 genome having a GC% content (54.1%) similar to that of its host species, *P. agglomerans* (55.3% ± 0.9%, based on the 16 complete genome assemblies available in the NCBI Genome database at the time of writing). The genomes of Nifs112 (50.2%) and Nufs112 (47.7%) deviated way more in that regard, although a *Pantoea*-infecting phage with a GC% content as low as 39% was previously documented (*Pantoea* phage vB_PagS_AAS21; MK770119.1; [22]).

Interestingly, the genome GC% of most of the *Pantoea agglomerans* phages uncovered so far is noticeably lower than that of their host (Table 1), hinting at the possibility of them having initially evolved to infect a different host and expanding/shifting their host range to *Pantoea* sp. only relatively recently on the evolutionary timescale if an assumption that phages try to adapt their GC% content to that of their hosts holds true. This observation yet again also highlighted that the GC% content of the phage should not necessarily be relatively similar to that of its host for successful, even though not automatically optimal, infection and replication. Previously, an analysis of more than 6000 bacterial chromosomes and more than 4500 complete phage genomes identified a linear relationship between the GC% content of phages and their hosts, with a linear regression equation that could explain ~81% of the variability showing that the phage genome GC% increases by ~0.88% for each 1% increase in the host GC content [34].

### 2.4. Scrutinizing the Place of Pantoea Phages Nifs112, Nufs112, and Nafs113 within the So-Far-Uncovered Phage Diversity

A BLASTN search against the viral sequences revealed that the degree of “novelty” (intergenomic distance to other cultured phages that had their genomes sequenced) possibly allows us to propose two of the studied phages (Nifs112 and Nafs113) as novel species within the already-recognized phage genera, whereas Nufs112 might even be a sole representative for a novel phage genus based on the ICTV adopted genome nucleotide sequence similarity phage genus and species demarcation criteria [35].

When querying the *Pantoea* phage Nifs112 (OK570184.1), the highest-total-scoring hit was documented to a complete genome of the *Erwinia* phage vB_EamP-S2, representing an *Eracentumvirus S2* species within the genus *Eracentumvirus* (NC_047917.1; 95% query coverage of 92.64% identity, resulting in ~88% intergenomic similarity corresponding to ~12% intergenomic distance). This hit to *Erwinia* phage vB_EamP-S2 was followed by hits to other *Erwinia* phages either officially or tentatively belonging to the *Eracentumvirus* genus (*Erwinia* phages Era103, Vyarbal, phiEa100, phiEa1H) among the top five highest-scoring hits.

In the case of the *Pantoea* phage Nufs112 (OK570185.1), the top hit to a cultured phage complete genome was noted to the *Klebsiella* phage 6939 (OL362271.1; 63% query coverage of 70.39% identity corresponding to ~55.7% intergenomic distance) unclassified at the genus and species level. This hit to *Klebsiella* phage 6939 was followed by hits to the phage Reminis, showing a similar intergenomic distance, and recognized or putative representatives of the *Gajwadongvirus* genus (*Escherichia* phage ECBP5, *Pseudomonas* phage MR4, *Pectobaterium* phage PP99), showing a >60% intergenomic distance to Nufs112.

Interestingly, the genome of the *Pantoea* phage Nafs113 (OK570186.1) had only a single meaningful hit—to *Pantoea* phage vB_PagM_LIET2 (NC_048751.1; 95% query coverage of 94.99% identity—~9.8% intergenomic distance) representing a sole *Lietduovirus LIET2* species within the genus *Lietduovirus*, with hits to other phage genomes having a ≤1 percent query coverage.

When the studied phage genomes were queried against the bacterial sequences found in GenBank (taxid: 2) using megablast, hits to bacterial genomes were limited to up to a few hundred nucleotides (less than a few percent query coverages) for each of the three phages, indicating a lack of highly similar prophages or the remnants of thereof in the so-far-sequenced bacterial isolates.

#### 2.4.1. Selected Protein Phylogenies

Having gained a glimpse at the genome nucleic acid sequence-wise similarity to other phages, a reconstruction of the selected analogous protein phylogenies was performed within the context of their most closely related counterparts encoded by other phage genomes (Figure 3). For this, four proteins with the same/similar annotation assigned in all three of the studied phages were selected with an assumption of a pairwise functional independence of the selected proteins for the lytic cycle of the phages (major capsid protein, terminase large subunit, endolysin, and phosphatase).

In all four of the trees, proteins from Nifs112 were most like their counterparts from the representatives of *Eracentumvirus*. Although the *Eracentumvirus* clade with Nifs112 had a well-supported MRCA in three of the trees (except the endolysin tree), its closest neighbors differed across all four of the trees with regard to taxonomical placement.

Selected proteins from Nafs113 shared an MRCA with the *Pantoea* phage vB_PagM_LIET2 belonging to the genus *Lietduovirus*, and in all of the trees save for an endolysin tree, there was a rather long branch leading to their two-sequence clade, indicating distinctness of their selected homologous proteins from the homologs found in other phages.

The situation with Nufs112 was revealed to be more ambiguous, as it did not reliably fall within the same monophyletic group, and the tree topologies surrounding the corresponding sequences of Nufs112 proteins differed to an extent. Based on the phylogenies, the selected Nufs112 proteins were most similar to the homologs from the *Klebsiella* phage 6939 and the *Ralstonia* phage Reminis. These two phages, however, shared a well-supported MRCA in all four of the trees, whereas Nufs112 reliably clustered with them only in the case of MCP and TerL. Interestingly, the phage Reminis and the *Pantoea* phage MR4 seemed to be classified in GenBank as tentative representatives of the genus *Gajwadongvirus*, but from the generated trees, only MR4, and not Reminis, seemed to form monophyletic clades together with the officially recognized Gajwadongviruses PP99 and ECBP5, which together boasted very short within-clade evolutionary distances. Thus, the designation of Reminis as a *Gajwadongvirus* representative did not really seem justified based on the selected protein phylogenies. While an evolutionary link to the recognized *Gajwadongvirus* proteins could also be established for Nufs112, their counterparts seemed to have diverged in a more distant past and were very unlikely to share an immediate ancestry.

#### 2.4.2. Proteome-Based Clustering

To get a better proteome-based overview of the place of the studied phages within the context of cultured and sequenced phage diversity, 18,553 complete phage genomes (including Nifs112, Nufs112, and Nafs113) available at INPHARED (1 October 2022 release; [36]) were subjected to vConTACT2 clustering [37]. Such an approach was chosen to benefit from a standardized re-annotation approach for all the genomes to be compared utilized by INPHARED, alleviating further interpretation of the results without having to worry about highly variable original submission genome annotation qualities.

From the vConTACT2-generated network, only the first neighbors of the studied phages were of interest, excluding the studied phages themselves—six unique neighbors were identified for Nafs113, nine for Nifs112, and 22 for Nufs112, whereas sixty-three of the phages were the first neighbors for both Nifs112 and Nufs112 (Appendix A). The studied phage first-neighbor sub-network (n = 103) was next visualized under an edge-weighted spring-embedded layout that placed more closely related phages (interconnected nodes) spatially closer to each other (Figure 4). Hosts of the identified first neighbors of the studied phages represented 17 bacterial genera, of which the most frequent one was *Escherichia* (35 phages), and other bacterial genera were represented by only up to eleven different phages infecting them. Apart from each other, there were no other first neighbors designated as *Pantoea* phages identified for either Nifs112 or Nufs112. However, three *Pantoea agglomerans* phages (LIET2, AAM37, and PSKM) were among the first neighbors of Nafs113, with the phage LIET2, expectedly from the previously determined BLASTN search, being the most similar one. Notably, VC_437_0, to which Nifs112 was clustered based on its proteome, otherwise comprised exclusively *Erwinia*-infecting phages, either officially or tentatively representing the genus *Eracentumvirus*.

Among the first neighbors of the studied phages, ninety-three represented the family *Autographiviridae*, with sixty-seven being further classified into the *Molineuxvirinae* subfamily comprising the clusters VC_436_0 (at least six genera representatives) and VC_437_0 (*Eracentumvirus*), and nineteen into the *Colwellvirinae* subfamily (VC_961_0; five different genera representatives). Interestingly, while fifty-eight *Molineuxvirinae* phages were the first neighbors of both Nifs112 and Nufs112 and only nine representatives of this subfamily were uniquely identified as the first neighbors of Nifs112, all nineteen *Colwellvirinae* representatives were neighbors of Nufs112 exclusively (Appendix A). Thus, it can be expected for Nufs112 to show some features of both *Colwellvirinae* and *Molineuxvirinae* phages, whereas Nifs112 is expected to have features characteristic of *Molineuxvirinae* phages. Nufs112, and seven more phages among its first neighbors, however, were classified as an overlap between VC_439 and VC_962, while representatives of neither of these two clusters were present in the first-neighbor-only network.

#### 2.4.3. Intergenomic Relationships within the Context of Proteomically Most-Similar Phages

Complete genomes of the phages comprising the studied phage first-neighbor subnetwork were also used as an input for VIRIDIC [38] to calculate their pairwise intergenomic sequence distances and subsequently perform a clustering at the ICTV set phage genus- and species-level intergenomic similarity demarcation criteria. The resulting distance matrix (Appendix A) was further visualized as an NJ tree and annotated (Figure 5).

Most of the genus-level designations in the retrieved complete-genome sequence-associated taxonomy coincided with the VIRIDIC clusterization at a 70% intergenomic similarity threshold, and such clades were monophyletic in the generated intergenomic distance NJ tree (Figure 5, e.g., *Acadevirus*, *Eracentumvirus*, *Kaohsiungvirus*, *Dibbivirus*, etc.). However, some of the (mostly tentative) designations in the taxonomy associated with the phage genome submissions were not supported at the chosen intergenomic distance criteria—*Vectrevirus* representatives were split into two VIRIDIC clades; *Uliginvirus* were split into five VIRIDIC clades at the given intergenomic similarity threshold; the bacteriophage Reminis was tentatively classified as *Gajwadongvirus* (likely incorrectly); the tentative *Tuodvirus Escherichia* phage vB_EcolP_P433.1 seemed way more similar to two clades, in which recognized or tentative *Vectrevirus* representatives are found, than to the officially recognized *Tuodvirus phD2B*.

In addition, it seemed very unbelievable to find a single alleged *Drexlerviridae* representative (*Vibrio* phage Vc1; KJ502657) to be so similar to a plethora of *Autographiviridae* podophages. A look at the ICTV virus metadata resource (VMR_20-190822_MSL37.3) revealed that this stemmed from the misattribution of a taxonomy, as the *Vibrio* phage Vc1 (KJ502657; family *Autographiviridae,* subfamily *Colwellvirinae*, genus *Gutovirus*) is a completely different phage to the *Vibrio* phage VC1 (MT360682; capitalized letter “C” in the phage name; genus *Jhansiroadvirus* in family *Drexlerviridae*), but the taxonomy associated with the latter was mistakenly expanded to the former in the GenBank.

Nevertheless, combining the results of the performed “overview” analyses for our studied phage place within the so-far-uncovered phage diversity, it was revealed that:Nifs112 was a representative of the genus *Eracentumvirus*, interestingly, so far comprising only *Erwinia* phages;Nufs112 was related to *Gajwadongvirus* representatives but did not look like a representative of that genus itself;Nafs113 could firmly be considered a representative of the genus *Lietduovirus*, but the question of whether it represents a novel species therein remains.

### 2.5. Differences between Nifs112, Nufs112, and Nafs113 and Their Most Closely Related Phage Relatives

#### 2.5.1. *Pantoea* Phage Nifs112

The complete genome of the *Pantoea* phage Nifs112 is a linear 46,202 bp long dsDNA molecule (including 296 bp short direct terminal repeats) with a GC% content of ~50.2%. The genome of Nifs112 (OK570184.1) was found to contain 59 open reading frames, all of which were located on a direct strand; no tRNA genes were found. ATG was predicted to serve as a start codon for fifty-one of the ORFs; GTG—four, CTG—two, and TTG–four ORFs. The majority of the predicted ORFs had a well-identifiable Shine–Dalgarno motif in the sequence span 20 bp upstream of the selected start codons (Appendix A). Taking similarity to the previously described phages and their proteins into account, we were able to reliably assign more than half (31/59) of the predicted ORF products with a more or less specific functional annotation. Most of the functionally annotated proteins Nifs112 encodes were responsible for virion morphogenesis. All four of the host lysis proteins expected in a phage infecting a Gram-negative host (holin, endolysin, inner and outer spanins) were identifiable in the genome of Nifs112. Additionally, DNA-directed RNA polymerase and several proteins responsible for DNA replication, modification, and repair were identified (e.g., DNA primase/helicase, DNA polymerase, DNA ligase, and several nucleases), as well as several proteins with a putative function that did not unambiguously fit any of the phage protein functional groups we defined (Appendix A).

As the results presented previously indicated that the *Pantoea* phage Nifs112 could be considered a novel representative of the *Eracentumvirus* phage genus, a closer genome architecture/proteome content comparison was next undertaken with exemplar isolates of both the *Eracentumvirus S2* and *Eracentumvirus era103* species recognized within the genus (*Erwinia* phage vB_EamP-S2 and *Erwinia* phage Era103, respectively; Figure 6; Appendix A).

Based on the Roary analysis at 30% identity, up to forty-two homologous proteins were encoded by all three (Nifs112, S2, Era103) of the phages (their total proteome comprised sixty-seven putative products at this threshold). Moreover, twenty of them were found in all three of the phages even at the 90% identity threshold set for Roary. All three of the phages shared a conserved genome architecture, with the main differences being in the presence or absence of short ORFs encoding different hypothetical proteins without an annotation. Pairwise homologous protein identity percentages clearly indicated that Nifs112 was more closely related to the phage vB_EamP-S2 than to Era103 (Figure 6).

Most of the ORF products of Nifs112 and vB_EamP-S2 were highly similar, showing exceptional conservation within the most functionally annotated products. The difference of interest, however, lay in the tailspike protein that was thought to contain an EPS depolymerase activity and showed a conservation only of about 50% between the two phages, with the sequence encoding the second half of the respective proteins being particularly different (Appendix A). Respective proteins from Nifs112 (UJH95830.1), vB_EamP-S2 (YP_009797658.1), and Era103 (YP_001039683.1) all had a cd20481-conserved domain (PSSMID 380473), representing the N-terminal and middle domains of a tailspike protein in the *Acinetobacter* bacteriophages identifiable using CD search, and *Acinetobacter* podophages are known to have a host-capsular-polysaccharide-degrading ability coupled to their tailspikes [27]. Previously, EPS activity was also documented in Era103 [39] and was proposed for vB_EamP-S2 [40]. Thus, while not looking very efficient based on the Nifs112 plaque appearance (Figure 2), Nifs112 seemed to also show depolymerase activity that was likely coupled to its tailspike protein.

Additionally, a region encoding the second hypothetical protein in the genome of the *Erwinia* phage vB_EamP-S2 (YP_009797612.1) seemed to be a fusion of two hypothetical proteins encoding ORFs found in both Nifs112 (UJH95776.1 from ORF4 (+2 frame) and UJH95777.1 from ORF5 (+3 frame)) and Era103 (YP_001039634.1 and YP_001039635.1) that were in close proximity of each other in both cases. The corresponding product/products so far seemed to be unique to *Eracentumvirus* representatives and remained without even a hint at the possible function. The ORF encoding nucleotidyl transferase in the genome of vB_EamP-S2 was also split in two in the genome of Nifs112, and it looked like it might now be translated with a −1 frameshift. Although these two ORFs overlapped, no evident slippery signal was found in the region of the overlap of ORF21 (+1 frame) and ORF22 (+3 frame) or ~60 bases upstream until the in-frame stop codon preceding the start codon of the ORF22, and the raw reads supported this sequence unambiguously. Moreover, both ORF21 and ORF22 had Shine–Dalgarno motifs that could reasonably well support the translation of their products (UJH95793.1 and UJH95794.1, respectively). However, given that most BLASTP-found homologs approximately correspond to their concatenated length, the question of whether such products would be functional is open.

Summing up the analyses, it seemed that the isolated and characterized *Pantoea* phage Nifs112 might not show a sufficient amount of similarity to representatives of either of the *Eracentumvirus S2* (*Erwinia* phages vB_EamP-S2 [40] and VyarbaL [41]) or *Eracentumvirus era103* (*Erwinia* phages Era103, phiEa1H, phiEa100 [42]) species and could be proposed as a sole representative of a novel species within that phage genus. As previously uncovered *Eracentumvirus* representatives are phages infecting *Erwinia*, it would be interesting to test how Nifs112 would fair against strains of *Erwinia*, as reports of phages capable of infecting representatives of both *Pantoea* and *Erwinia* were previously published [43], which, in the case of the successful ability to lyse *Erwinia*, would raise particular interest for further studies of Nifs112 as a phytopathogenic bacteria biocontrol agent. The hypothesis that *P. agglomerans* LS5-2 (the isolation host of Nifs112) might not be an optimal host for Nifs112 could further be reasoned by the fact that, for example, the closely related *Erwinia* phage VyarbaL formed plaques of 3–3.5 mm in diameter and a well-marked halo on its host strain of *E. amylovora* (whereas Nifs112 plaques on its isolation host *P. agglomerans* strain LS5-2 were way smaller and with a very opaque halo, although the plaque size is usually mostly dependent on the percentage of agar in the top layer “soft” medium, which was not mentioned in case of preprint describing VyarbaL) [41]. Additionally, the *Erwinia* phage S2, representing the same genus *Eracentumvirus* species as VyarbaL, was able to lyse not only multiple strains of *E. amylovora* but also at least a single strain of both *P. agglomerans* (strain Em283) and *P. ananatis* (strain 351 Lys) [44].

#### 2.5.2. *Pantoea* Phage Nufs112

The complete genome of the *Pantoea* phage Nufs112 is a linear 45,951 bp long dsDNA molecule (including 410 bp short direct terminal repeats) with a GC% content of ~47.7%. The genome of Nufs112 (OK570185.1) was predicted to have 67 open reading frames and no tRNA genes; every ORF was detected on a direct strand. ATG was predicted to be a start codon for fifty-eight of the ORFs, TTG for 6, GTG for 3, and no ORFs were presumably starting with a CTG. Only several of the identified ORFs did not have a highly preserved Shine–Dalgarno motif complementary to the 16S rRNA tail of *P. agglomerans* in the sequence span 20 bp upstream of the selected start codons (Appendix A).

After thorough functional annotation of the predicted ORF products, we were able to assign a function to slightly less than half of them (32/67). Nearly all of the functionally annotated ORF products of Nufs112 were revealed to perform either analogous or similar functions to those identified for Nifs112 products and are typically expected to be present in an *Autographiviridae* phage. The functionally annotated proteins from the Nufs112 proteome included a DNA-directed RNA polymerase, all the envisaged proteins involved in virion morphogenesis (e.g., MCP, portal, TerL, TerS, a few internal virion proteins typical to *Autographivirdae* representatives [45]), and several tail proteins), Gram-negative host phage lysis proteins (holin, endolysin, and inner and outer spanins), and characteristic proteins responsible for DNA replication, modification, and repair (e.g., DNA primase/helicase, DNA polymerase, DNA ligase, and several nucleases). Of course, some proteins with a putative function that did not fall within any of the phage protein functional groups we defined were also present (Appendix A).

The data presented previously indicated that the *Pantoea* phage Nufs112 was most closely related to the *Klebsiella* phage 6939 and the phage Reminis (which, based on the submission-associated taxonomy, was proposed to be included in the *Gajwadongvirus* genus without any justification). From the taxonomically officially recognized phages, Nufs112, indeed, was so far most similar to the representatives of the *Gajwadongvirus* phage genus (*Escherichia* phage ECBP5 and *Pectobacterium* phage PP99).

In the Roary analysis with a 30% protein identity threshold, the panproteome of these five phages (Nufs112, 6939, Reminis, ECBP5, PP99) comprised one hundred and thirty-five products, with twenty-nine of them being encoded by the genome of each phage in the comparison. At this threshold, Nufs112 encoded up to twenty-one proteins not found in the proteomes of its closest relatives, nineteen of which remained without any functional annotation (the other two being HNH endonuclease encoded by ORF31, and a C39 family peptidase—a product of ORF66), whereas at the 90% threshold there was no protein shared between the five phages, which was not very surprising given their large pairwise intergenomic distances. The *Pantoea* phage Nufs112 only shared a hypothetical product of ORF67 with Reminis under such a strict threshold. Notably, it was shown that both the recognized *Gajwadongvirus* species (represented by ECBP5 and PP99) shared up to 32 of the proteins, having a >90% pairwise identity, whereas Reminis had only a single hypothetical protein shared with a recognized representative of *Gajwadongvirus* at such a threshold (Appendix A). In line with previous analyses, Reminis was shown to be more related to a taxonomically unrecognized *Klebsiella* phage 6939 rather than *Gajwadongvirus* representatives and could not be considered a tentative *Gajwadongvirus*.

For the comparison of the genome architectures of the *Pantoea* phage Nufs112 (OK570185) and closely related phages, genomes of the *Ralstonia* phage Reminis (MN478376) and the *Klebsiella* phage 6939 (OL362271) were rearranged to ensure collinearity with Nufs112, for which the exact genome termini were identified. In addition, it was revealed that all of the five genomes should be colinear, as seen within the virions of the respective phages. Consistent with the expectations based on the results described previously, nearly all of the proteins found in Nufs112 that were also present in the *Klebsiella* phage 6939 and the *Ralstonia* phage Reminis, and the *Gajwadongvirus* representatives (*Escherichia* phage ECBP5 (KJ749827; [46]) and the *Pectobacterium* phage PP99 (NC_047802; [47])), were more similar to their counterparts in phage 6939 and Reminis (Figure 7; Appendix A).

As in the case of the Nifs112 comparison with *Eracentumvirus* representatives, the genomes of Nufs112 and its closest relatives also had a well-defined modular structure (Figure 7, Appendix A). The genome of Nufs112 began with a 4.8 kbp stretch containing up to 14 putative ORFs encoding hypothetical proteins, which was followed by an ORF encoding a hallmark of *Autographiviridae* podophages–DNA-directed RNA polymerase (ORF15; UJH95846.1). The region from 7.5 kbp up to ~23 kbp contained ORFs encoding proteins responsible for DNA replication, modification, and repair, as well as several additional functions not reliably falling within any of the defined protein functional groups. Functionally annotated proteins encoding ORFs in this part of the genome were interspaced by up to another 16 hypothetical proteins encoding ORFs. The remainder of the genome (~23 kbp to ~46 kbp) was mostly comprised of ORFs encoding virion morphogenesis and putative host lysis proteins that did not form a lysis cassette with sequential ORFs encoding up to four of the lysis proteins (holin, endolysin, and inner and outer spanins) but were interspaced by other morphogenesis proteins coding ORFs.

The main difference that stood out when comparing the proteome contents of these phages was, however, a putative tail fiber encoding ORF (ORF52 in Nufs112, UJH95883.1).These ORF products all had an identifiable T7 tail-fiber-protein conserved domain (pfam03906) at their N termini but differed greatly in lengths (242 aa in Reminis (QGH45085.1); 306 aa in Gajwadongviruses PP99 (YP_009788795.1) and ECBP5 (AID17699.1); 506 aa in Nufs112 (UJH95883.1); 769 aa in *Klebsiella* phage 6939 (URY99237.1) and pairwise similarities. Considering that tail fiber/fibers (their sequence) is one of the factors influencing the host range of phages, it is not uncommon for phages of different hosts to have these analogous proteins be diverged. However, it is worth noting that the Gajwadongviruses ECBP5, PP99, and MR4 (which is currently only a tentative *Gajwadongvirus* representative, but rightfully so) were all isolated on different hosts (*Escherichia*, *Pectobacterium*, and *Pseudomonas* sp., respectively), despite having very similar putative tail fiber proteins (pairwise identity >90% over 306 aa length of the respective proteins).

The predicted tailspike protein of Nufs112 (product of ORF64, UJH95895.1) had an identifiable pectate lyase superfamily protein domain (pfam12708), which was previously also identified in other phages [48]. The activity of this protein, a multitude of extra copies of which are expected to accumulate during Nufs112 progeny virion morphogenesis within the host and then be released into the surrounding environment alongside phage progeny, might be an explanation for the halo zones observed around the Nufs112 plaques (Figure 2). Such an individual protein showing depolymerase activity is expected to diffuse way further than the virions of the phage itself in a double agar overlay, and might be considered a candidate for further studies evaluating the product of Nufs112 ORF64 as an enzymatical antibacterial itself [48].

Consolidating the results, the *Pantoea* phage Nufs112 was sufficiently divergent from any of the so-far-uncovered phages, which allows us to propose it to be the founding member of a novel phage genus within the family *Autographiviridae*.

Of note, during the preparation of the genome architecture comparison figure between the phages related to Nufs112, and other analyses performed within this study, several discrepancies between the results presented in the article describing the phage Reminis [49] and referring to its genome (MN478376) were documented (e.g., no *Ralstonia solanacearum* was indicated as the host in the metadata of MN478376, the genome of Reminis (MN478376) was not organized taking the SDTRs of 377 bp mentioned by the authors into account, and no SDTRs were present and/or annotated in MN478376, proteome contents and evolutionary analyses using MN478376 indicated that it is a podophage, and not a siphophage, among others).

#### 2.5.3. *Pantoea* Phage Nafs113

The complete genome of the *Pantoea* phage Nafs113 is a linear 75,899 bp long dsDNA molecule with a GC% content of ~54.1%. No defined termini could be predicted as an implication of the presumed headful packaging strategy that results in the progeny virions each having terminally redundant genomes that are circularly permuted when compared to each other. The scaffold representing the genome of Nafs113 was opened in a way that ensured collinearity with the closest relative, which in this particular case also followed the convention of cutting the circularized scaffold before ORFs encoding terminase subunits. Although, frankly, in the case of both phages, at the time there seemed to be no reason to assume that the ORF preceding TerL-encoding ORF encodes TerS, which is a protein known not to show such a high degree of evolutionary conservation as its larger counterpart, other than empirical phage genomics observations of the common TerS- and TerL-encoding ORF appearance in tandem. However, it is worth noting that with the growth of the number of phage genome sequences in the public biological sequence repositories, more and more exceptions to most “phage genomics rules” begin to appear.

The genome of Nafs113 (OK570186.1) was found to contain 130 open reading frames, 43 of which were located on a direct strand, whereas 87 were located on a reverse strand. No tRNA genes were found in the genome of Nafs113. ATG was predicted to serve as a start codon for one hundred and sixteen of the ORFs, GTG for nine, TTG for four ORFs, and a single ORF was predicted to have CTG as a start codon.

Interestingly, despite the fact that most of the predicted ORFs had a well-identifiable Shine–Dalgarno motif in the sequence span 20 bp upstream of the selected start codons (Appendix A), the ones having subpar or “weak” possible ribosome-binding sites were also abundant, especially among the ORFs putatively encoding hypothetical proteins, and these “weak” predicted SD motifs were much more frequent than in the genomes of Nifs112 and Nufs112. For example, the average DeltaG (a change in the free energy (kcal/mol) required to bring the two strands of nucleotides (putative SD sequence-containing sequence upstream of studied phage ORF and *P. agglomerans* antiSD sequence)) of Nafs113 was only −4.57 ± 2.36 kcal/mol (average ± SD) as calculated by free_align.pl script [50], whereas the Nifs112 and Nufs112 ORFs could be presumed to allow binding of their putative SD sequences and antiSD sequence of the host ribosomes much better on average, showing a lower DeltaG of −6.05 ± 2.72 kcal/mol and −6.82 ± 3.14 kcal/mol for Nifs112 and Nufs112, respectively.

As the *Pantoea* phage Nafs113 was demonstrated to be rather unique among the so-far publicly available sequenced phages (bar the *Pantoea* phage vB_PagM_LIET2) based on its complete-genome nucleotide sequence, it was not unexpected that only 30% of the predicted Nafs113 ORF products could be assigned a putative function. Nevertheless, a number of the reasonably well-conserved proteins responsible for virion morphogenesis/structural features expected for a tailed dsDNA phage were identified in the genome of Nafs113 despite its genomic nucleotide sequence being very uncommon among the so-far-uncovered phage diversity. Both tail-tape-measure-protein- and tail-sheath-protein-encoding ORFs were identified, which would be sufficient to identify Nafs113 as a myophage had the TEM virion analysis not yet been made. Upon closer inspection, holin, endolysin, and putative inner and outer spanins, representing the lysis proteins expected for a phage of a Gram-negative host bacterium, could be identified. In addition, a number of nucleases and other proteins involved in DNA replication, modification, and repair were annotated. The genome of Nafs113 contained several additional functions performing proteins as well; however, their annotations did not allow us to reliably categorize them into any of the phage product groups we categorized proteins into according to their function (Appendix A; Figure 8).

As was understood up to this point, the *Pantoea* phage Nafs113 so far had only a single meaningful phage relative sequenced, although a very close one—*Pantoea* phage LIET2—so far a single representative of the sole *Lietduovirus* genus species *Lietduovirus LIET2*. Undeniably, previous results warrant Nafs113 a place within the same *Lietduovirus* genus, but does it represent a novel species within it?

The genomes of Nafs113 and LIET2 boasted the same genome architecture, with genes involved in particle morphogenesis being located in at least two separate distinguishable “capsid” and “tail” gene modules separated by a stretch of several tens of other ORFs, and the TerL encoding ORF not being in close proximity of either. ORFs encoding proteins involved in host lysis were also scattered throughout three locations in the genome, and the putative inner and outer spanin ORFs appeared in tandem (Figure 8; Appendix A). Roary identified that both phages shared 120 genes/proteins at a 30% identity threshold (and still 100 of them under a rather conservative threshold of 90% identity). This was especially striking, given that their predicted panproteome comprised only 141 proteins at 30% identity. At such a low similarity threshold, all eleven proteins distinctive to LIET2 but not Nafs113 were without a functional assignment (“hypothetical proteins”), while those exclusive to Nafs113 included two endonucleases (ORF60 and ORF115 products) among the proteins without a functional assignment. These distinct segments between the genomes of Nafs113 and LIET2 were clearly visible when comparing the genome architectures and either the underlying ORF product amino acid sequences (Figure 8) or the respective genomic region nucleotide sequences (Appendix A).

Of course, based on previous observations regarding the unusual putative SD sequences of Nafs113 detected during the annotation, we decided to look at the putative SD sequences 20 bp upstream of the vB_PagM_LIET2 ORFs as seen in the annotations of NC_048751.1. Shockingly, even a proportionally higher amount of subpar “weak” SD sequences was identified, with the DeltaG of LIET2 ORFs averaging at −4.14 ± 2.74 kcal/mol, which was more than the average for the Nafs113 ORFs. As LIET2 and Nafs113 were annotated independently of each other, the unusually high occurrence of the atypical, seemingly subpar, SD sequences seemed to be a genomic feature of the *Lietduovirus* phage genus representatives and is worth being investigated further.

Another marked difference between the two genomes was that one of the two ORFs that were either fused into one (LIET2) or split into two (Nafs113) encoded what was annotated as a 1218 aa long EPS depolymerase-domain-containing protein in LIET2 (YP_009843765.1) or two putative tail-spike (UJH95983.1 and UJH95984.1)-encoding ORFs in Nafs113. While a CD search failed to identify any conserved domains that would imply EPS depolymerase activity for either of these three proteins, the top HHpred hit for each of the three of them was to a PDB structure (6TGF_E; probabilities ≥ 99.92, e-values ≤ 4.5 × 10^−24^), representing a *Pantoea stewartii* WceF—a glycan biofilm-modifying enzyme with a bacteriophage tailspike-like parallel beta-helix fold. It was recently shown that WceF is a glycosidase active on stewartan, which acts as the main *P. stewartii* EPS biofilm component, and that WceF is very similar to bacteriophage tailspike proteins [51]. The same authors also noted that WceF homologs are also present in other plant pathogens of bacterial origin, but the functional role of these proteins is so far not fully understood [51]. Although ORFs encoding these products overlapped in the genome of Nafs113 by about 30 bp, they were in +1 and +3 reading frames, respectively. When comparing their concatenated amino acid sequence with the sequence of the respective LIET2 protein, they were nearly identical, and the only gaps identifiable in their alignment are the last nine amino acids of the Nafs113 ORF85 product, before the perfect match in the region corresponding to the N terminus of the protein encoded by ORF86 (apart from the first amino acid of the protein). Interestingly, no evident −1 signal for a possible frameshift was identified, and the underlying nucleotide sequence of the region was unambiguously supported by the raw reads. Given that the presumed SD motif for translation of ORF86 was rather weak, especially in comparison to that of the ORF85, we hypothesized that the truncated form of the EPS depolymerase-domain-containing protein found in LIET2 (YP_009843765.1) corresponding to the product of ORF85 of Nafs113 was enough for a successful lytic cycle on its host strain, while the production of the C terminal part corresponding to the product of ORF86 might be significantly hampered given the genomic context in Nafs113 in the case of an absence of the ribosomal frameshift during translation of ORF85. However, as was noted earlier, quite a lot of ORFs in the genomes of both Nafs113 and LIET2 had rather “weak” putative SD sequences upstream of their predicted start codons, and further studies would be necessary to understand its effects on the translation of the respective products. As the product/products of these ORFs are believed to be structural, further studies (e.g., SDS-PAGE and mass spectrometry analysis) would be interesting to check whether the two proteins (ORF85 and ORF86 products) are really produced individually, or some kind of a less-evident frameshift mechanism resulting in a concatenated form is present.

A similar fusion/split was observed for two Nafs113 hypothetical proteins (UJH96005.1 and UJH96006.1) encoded, this time, from frames −1 (ORF107) and −3 (ORF108) in Nafs113, respectively, whose products appeared to be fused in LIET2 (YP_009843788.1). Again, in this case, too, no apparent +1 frameshift signal, was observed, and the presumed Shine–Dalgarno motif for translation of the ORF107 product appeared “stronger” than that of ORF108. However, in this case, the ORF107 product (UJH96005.1) had a lot of BLASTP hits to hypothetical proteins of similar length (~208 aa) from the bacteria (query coverage of ~99% and identities around 50%, e-values less than 1 × 10^−35^), whereas the same search using the amino acid sequence of the putative ORF108 product (UJH96005.1) as a query revealed only a match to the hypothetical protein HWC07_gp107 from the *Pantoea* phage vB_PagM_LIET2.

Based on the phage LIET2 description available in the Master thesis of Emilija Petrauskaitė [22], LIET2 virions had exactly the same morphology and, probably, in reality, indistinguishable dimensions (heads 114.9 ± 9.6 nm long and 44.6 ± 5.4 nm wide, contractile tails 87.8 ± 9.2 nm in length) from those of Nafs113, measurements of which were presented earlier in the Results and Discussion Section 2.2 (Figure 2, bottom row). The difference was noted in plaque sizes, as, like Nafs113, LIET2 seemed to form transparent plaques without a halo around them, but the plaques of LIET2 were claimed to be 0.65 mm in diameter on average (although they seemed to reach diameters of slightly more than 1 mm from Figure 5D in the corresponding thesis [22]) in contrast to the plaques of Nafs113 that could reach even 2 mm in diameter, although on average being around ~1 mm as well when using soft LB agar with 0.7% agar concentration for double agar overlays (Figure 2, top row). However, a plaque size comparison such as that is highly unreliable, as there are a plethora of other factors that might influence the plaque size such as host strain, host culture freshness, agars used, etc., which violates the ceteris paribus principle, under which only the phage added in a double agar overlay should be different for a maximally correct comparison of different phage negative colonies in any respect. Interestingly, despite hints at the putative EPS depolymerase activity of the LIET2 tailspike protein, which seemed to be in its intact form—encoded from a single ORF (e.g., not split into two ORFs as in the case of Nafs113, which did not form haloes around plaques)—the plaques of LIET2 still did not seem to have haloes around them (Figure 5D in [22]).

All things considered, there is undeniably a very close evolutionary relationship between the *Pantoea* phage LIET2 and the *Pantoea* phage Nafs113 that unanimously allowed us to propose the classification of Nafs113 as a representative of the *Lietduovirus* genus. We find it fascinating that LIET2 and Nafs113 were both isolated from the source material (Nafs113—insect-associated source; LIET2—plant-associated source, both of which are still extremely interconnected ecosystem-wise) collected geographically (Nafs113—Latvia; LIET2—Lithuania) close and temporally proximal (Nafs113—2008; LIET2—2015) to each other. Speculatively, it can be proposed that the consequences of differences in horizontal gene exchange events by their ancestral phage populations (e.g., acquisition of different auxiliary genes) might explain most of the differences observed between genome architectures and proteome contents as, we believe, the ecological niches and environmental factors naturally influencing both LIET2 and Nafs113, as well as their respective host species strains, might naturally be very similar.

Given their great difference from any other phage sequenced so far, at this time, we are rather convinced that, for practical reasons, even though intergenomic distances between *Pantoea* phages LIET2 and Nafs113 were higher than the same phage species level threshold of <5% differences over complete genome lengths, and some genomic features were different, there is no sufficient evidence in favor of creating a novel phage species within the genus *Lietduovirus* for Nafs113 instead of considering it another isolate representing the *Lietduovirus LIET2* species. Additional research on both of these phages might, however, shed light on more of their functional differences that may warrant their possible delineation at the species level in the future.

## 3. Materials and Methods

### 3.1. Host Bacteria and Phage Isolation and Propagation

Studied phages were isolated alongside their host bacteria from a dead unidentified Latvian grasshopper (order *Orthoptera*) specimen, as described previously for the phage Nocturne116 [52]. One of the several indicator cultures obtained from the suspension of the crushed insect, namely LS5-2, which formed yellow-colored colonies after incubation for 48 h at room temperature (RT) and was able to grow at +30 °C as well, was chosen for the subsequent “phage hunting”.

A total of 50 μL of LS5-2 indicator culture was mixed with 50–100 μL aliquots of crushed grasshopper suspension, filtered through a 0.45 μm pore size syringe filter (Sarstedt, Nümbrecht, Germany), and subjected to a double agar overlay assay [53]. Phage-negative colonies of several different appearances were observed on a lawn of bacterial culture, LS5-2, after incubation for 24 h at RT. Although several different agarized media were used when working with these phages, initially, LB (g/L: Bacto-Tryptone—10, Bacto yeast extract—5, NaCl—10) medium was used with the addition of 15 g/L agar for the bottom layer and 7 g/L agar for the top “soft” layer.

Several rounds of individual plaque purifications were carried out, and three different lines of plaques consistently giving the same negative colony appearances were established and assumed to represent three distinct phages.

### 3.2. Phage Concentration and Purification

For phage propagation, soft agar layers from 20 double agar overlay plates (for each phage individually) showing confluent lysis of the bacterial lawn were collected and homogenized by vortexing with the addition of 5 mL LB liquid medium per plate. Next, the homogenized soft agar layer mixture was subjected to centrifugation (12,000× *g*, 30 min), and the resulting supernatant was decanted and filtered through a 0.45 μm pore size syringe filter. For all three of the investigated phages, such propagation yielded filtrate with a titer of up to 1–2 × 10^10^ PFU/mL.

Phages were then concentrated by centrifugation of the filtrates in a Beckman Optima L-100XP ultracentrifuge with a 70 Ti rotor (48,000× *g*, total centrifugation time 1 h at +4 °C; Beckman Coulter, Brea, CA, USA). The obtained phage pellets were dissolved in ~4 mL of TE buffer (10 mM Tris, 1 mM EDTA, pH 8.0). Acquired phage suspensions were next layered on top of a CsCl solution (CsCl—0.6 g per mL of the TE buffer). For each phage, a single Ultra-Clear centrifuge tube (14 × 95 mm, Beckman Coulter) was filled with 11.5 mL of the CsCl solution, and 2 mL of a concentrated phage sample was loaded on top of the CsCl solution. Solutions were then spun at +4 °C for 20 h (24,000 rpm; 100,000× *g* max) in an SW 40 Ti rotor (Beckman Coulter)-equipped Beckman Optima L-100XP ultracentrifuge. Distinct horizontal phage bands were formed for each of the phages, and these were collected by pipetting. Collected phage-containing bands were desalted using NAP-25/Sephadex G-25 columns (Pharmacia, Uppsala, Sweden) and PBS as an exchange buffer.

### 3.3. Phage Sample Transmission Electron Microscopy and Virion Dimension Determination

To make studied phage virion micrographs, 5 μL of the purified phage specimens were allowed to adsorb onto carbonized formvar-coated 300 mesh copper grids (Agar Scientific, Stansted, UK) for 5 min, rinsed with 1 mM EDTA solution, and negatively stained using 0.5% uranyl acetate before being left to dry for 2 h. After drying, the stained phage-sample-containing grids were examined using a JEM-1230 transmission electron microscope (accelerating voltage of 100 kV was used; JEOL, Akishima, Japan), and virions were pictured with a Morada 11 MegaPixel TEM CCD microscope-mounted camera using iTEM imaging software (Olympus, Tokyo, Japan).

Phage particle dimensions were determined with the help of ImageJ software (v1.52a; [54]) utilities using micrograph scale bars for pixel-to-nm ratios. Different micrographs depicting presumably intact zoomed-in virions were used to measure capsid lengths and widths, as well as tail lengths, for each of the isolated phages. Phage particle dimensions presented within this study represent an average ± standard deviation for the corresponding virion structural features measured from at least six independent micrographs taken at different fields of view (for each of the three studied phages).

### 3.4. Phage Genomic DNA Extraction and Whole-Genome Sequencing

To release the phage genomic DNA, the purified and concentrated phage sample was incubated at +56 °C for 1 h with the addition of proteinase K and SDS (0.5% final concentration). DNA extraction was performed with the Genomic DNA Clean & Concentrator-10 (Zymo Research, Irvine, CA, USA) kit as per the manufacturer’s protocol. Quality and approximate concentration of the obtained phage genomic DNA were evaluated on a NanoDrop ND—1000 spectrophotometer (Thermo Fisher Scientific, Thermo Fisher Scientific, Waltham, MA, USA) and then diluted accordingly to verify the specific dsDNA amount using a Qubit fluorometer (Invitrogen, Waltham, MA, USA) dsDNA high-sensitivity quantification assay (Invitrogen).

The TruSeq DNA Nano Low Throughput Library Prep Kit (Illumina, San Diego, CA, USA) protocol, compatible with the TruSeq DNA Single Indices Set A (Illumina), was followed to prepare the Illumina MiSeq-compatible DNA libraries (different adapters were used for each of the phages to allow further pooling and unambiguous demultiplexing of the data). To prepare the appropriate input for the library preparation step, 200 ng of the dsDNA from each of the studied phages was randomly fragmented with a target fragment length of 550 bp in mind using a Covaris S220 focused ultrasonicator (Covaris, Woburn, MA, USA).

Resultant library quality control was performed using an Agilent 2100 bioanalyzer (Agilent, Santa Clara, CA, USA) with a high-sensitivity DNA kit (Agilent) and a Qubit fluorometer (Invitrogen) dsDNA high-sensitivity quantification assay (Invitrogen). Libraries were then sequenced on the Illumina MiSeq system (Illumina) using a 500-cycle MiSeq Reagent Kit v2 nano (Illumina) for three of the twelve uniquely indexed libraries comprising the run.

### 3.5. Taxonomic Identification of the Host

Overnight culture of the bacterial host isolate LS5-2 was used for genomic DNA isolation as described for the phage genomic DNA extraction (first paragraph of Section 2.4). Polymerase chain reaction using universal 27F and 1492R primers ([55]; ordered at Metabion, Steinkirchen, Germany) and DreamTaq Hot Start DNA polymerase (Thermo Fisher Scientific) was carried out (program: 3 min at +95 °C, 35 cycles of (30 s at +95 °C, 30 s at +55 °C, 1.5 min at +72 °C), 10 min at +72 °C, followed by a hold at +4 °C), and the PCR product was subjected to native agarose gel electrophoresis. A band of interest (~1450 bp) was extracted from the gel using the GeneJET Gel Extraction Kit (Thermo Fisher Scientific) according to the manufacturer’s guidelines.

Sanger-based sequencing of the extracted amplified partial 16S rRNA sequence of the host was carried out in two reactions (27F primer for forward and 1492R primer for reverse read) according to the BigDye^®^ Terminator v3.1 Cycle Sequencing Kit’s (Applied Biosystems, Waltham, MA, USA) recommendations (program: 3 min at +96 °C, 30 cycles of (10 s at +96 °C, 5 s at +50 °C, 4 min at +60 °C), 5 min at +70 °C, followed by a hold at +4 °C). The ABI PRISM 3130xl system (Thermo Fisher Scientific) was used as a sequencer. Raw read chromatograms were manually inspected and trimmed in GeneStudio (v. 2.2.0.0.). After, the trimmed reads were joined into a contig and used for consensus sequence calling (higher-quality traces were used in case of ambiguities in the read overlap region) in the same software.

The resulting partial 16S rRNA gene sequence of the host was queried against the bacterial 16S rRNA gene sequences publicly available at EzBioCloud [56]. All valid name hits to the query sequence were retrieved from EzBioCloud in the form of sequences, alongside the 16S rRNA gene sequence from the *Pectobacterium carotovorum* strain NCPPB 312 (JQHJ01000001) representing an outgroup. Multiple sequence alignment (MSA) was performed using MAFFT (v. 7.453; [57]), and the resulting MSA was used for Neigbor-Joining (NJ) tree [58] reconstruction in MEGA7 [59]. The NJ tree was built using the p-distance method, including both transitions and transversions, assuming uniform rates among sites and homogeneous patterns among lineages, employing a 90% site coverage cut-off for the input alignment. Branch supports were assessed using 1000 bootstrap replicates [60]. The resulting tree was outgroup rooted and visualized in FigTree (Rambaut, A. FigTree v. 1.4.4; available online: http://tree.bio.ed.ac.uk/software/figtree/ accessed on 10 May 2021) with further manual visual refinement in Inkscape (v. 1.0.1; available online: https://inkscape.org accessed on 10 May 2021).

### 3.6. Phage Genome De Novo Assembly, Functional Annotation, and Termini Verification

Prior to de novo assembly, demultiplexed read datasets were inspected with the help of FastQC (v. 0.11.9; [61]), and corresponding library reads were trimmed using bbduk script from the bbmap (v. 38.41) package (any remaining adapters, bases below 20 Phred quality, and reads shorter than 50 bp were removed [62]). Trimmed reads were then used as an input for Unicycler (v. 0.4.8; [63]) in “normal” mode.

De novo assembled “circularized” contigs were next used alongside the corresponding library untrimmed reads for possible genome termini identification using PhageTerm (v. 1.0.12; [64]). Next, raw reads were mapped on reorganized contigs (BWA-MEM v. 0.7.17-r1188; [65]), and the sequence alignment maps were manually inspected in UGENE (v. 37.0; [66]). Custom primers (Table 3) were designed for run-off Sanger-based sequencing to test the predicted genome termini, where a signal for a defined genome termini was documented by PhageTerm.

The open reading frame (ORF) and tRNA gene predictions were performed with the corresponding tool implementations in DNA master (v 5.23.6; https://phagesdb.org/DNAMaster/ accessed on 12 April 2021) and were followed by putative ORF product functional annotations using publicly available biological sequence repositories and web services as described previously [52]. However, this time, 13 bases of the *P. agglomerans* 16S rRNA tail (3′-AUUCCUCCACUAG-5′; [67]) were used for the free_align.pl script [50] when evaluating the presence of putative Shine–Dalgarno (SD) sequences in regions 20 bases upstream of the predicted ORF putative start codons within the genomes of the studied *Pantoea* phages (free_align.pl -o “20 bp upstream of the start codon” AUUCCUCCACUAG; G-U wobble base pairs allowed; putative SD sequences with Delta-G of more than −3.60 kcal/mol at +37 °C binding temperature were considered subpar). Longer possible coding sequence overlaps were also evaluated when annotating phages within this study. Additionally, TMHMM [68] and Phobius [69] web servers were used when looking for putative lysis module proteins.

To further verify the packaging strategies/termini types the isolated phages employed, the terminase large subunit (TerL) amino acid sequence ML tree was reconstructed using an extended dataset of Merrill and colleagues [30]. Extension of the dataset included two sequences for an SPO1-type long direct terminal repeat clade (that of *Bacillus* phage BJ4 (AOZ61694) and *Bacillus* phage SBP8a (AOZ62321), as well as the TerL sequences of the studied *Pantoea* phages Nufs112 (UJH95886.1), Nifs112 (UJH95823.1), and Nafs113 (UJH95900.1)). Multiple TerL sequence alignment was performed using MAFFT (v. 7.453; [57]), and ML phylogeny was reconstructed in IQ-TREE (v. 2.0.6; [70]), selecting the best-fit substitution model based on the ModelFinder [71] inference, allowing for polytomies and evaluating branch supports using 1000 ultrafast bootstrap (UFBoot; [72]) replicates. The resulting tree was midpoint rooted and visualized in FigTree (Rambaut, A. FigTree v. 1.4.4; available online: http://tree.bio.ed.ac.uk/software/figtree/ accessed on 10 May 2021).

### 3.7. Elucidating the Place of Isolated Phages within the So-Far-Uncovered Phage Diversity

First, a BLASTN search against all the viral sequences (taxid:10239) publicly available at the non-redundant nucleotide collection was performed for complete-genome sequences of studied phages to evaluate their novelty and get a hint at the most appropriate further analyses for the task of elucidating the place of isolated phages within the so-far-uncovered phage diversity.

#### 3.7.1. Proteome-Based Clustering with Other Cultured Phages

As complete annotated genomes of *Pantoea* phages Nifs112 (OK570184.1), Nufs112 (OK570185.1), and Nafs113 (OK570186.1) were already publicly available in GenBank in late January 2022, we took advantage of the INPHARED [36] database utilities (1 October 2022 release) and proceeded with a standard vConTACT2 (v. 0.11.3; [37]) proteome-based clustering of the publicly available complete phage genomes (n = 18,553) using relevant INPHARED inputs. The vConTACT2-generated protein-sharing network was visualized and annotated in Cytoscape (v. 3.8.2; [73]). Nodes representing the studied phages were selected along with their first neighbors, and a new network was created from the selection, which was further visualized in an edge-weighted spring-embedded layout and annotated.

#### 3.7.2. Intergenomic Distances to Other Cultured Phages

The vConTACT2-identified first-neighbor genomes were next subjected to VIRIDIC (v. 1.0; [38]) analysis alongside the studied phages (n = 100 + 3) to calculate pairwise intergenomic distances between them under default settings. The resulting intergenomic distance matrix was then used to create a neighbor-joining tree with the help of the package “ape”(v. 5.5; [74]) in R. The resulting tree was then midpoint rooted, visualized, and annotated with the help of FigTree (Rambaut, A. FigTree v. 1.4.4; available online: http://tree.bio.ed.ac.uk/software/figtree/ accessed on 10 May 2021) and Inkscape (v. 1.0.1; available online: https://inkscape.org accessed on 10 May 2021).

#### 3.7.3. Selected Studied Phage Protein Phylogeny Reconstruction

Individual protein phylogenies were reconstructed for the major capsid protein (UJH95814.1; UJH95877.1; UJH95932.1), terminase large subunit (UJH95823.1; UJH95886.1; UJH95900.1), endolysin (UJH95826.1; UJH95889.1; UJH95998.1), and phosphatase (UJH95805.1; UJH95868.1; UJH95946.1) of the studied phages (accessions given for Nifs112, Nufs112, and Nafs113, respectively).

First, these sequences were subjected to a BLASTP search against the non-redundant protein sequences of viral origin (taxid:10239) under other default settings. For a given protein from the studied phages, the ten highest-scoring hits encoded by other cultured phage genomes were selected and downloaded regardless of the annotations, combining hits into multifastas comprising proteins of presumably analogous functions.

Next, respective analogous protein amino-acid-sequence-containing datasets (proteins from Nifs112, Nufs112, Nafs113, and ten highest scoring hits from other phages for each) were aligned using MAFFT (v. 7.453; [57]) and subjected to ML phylogeny reconstruction in IQ-TREE (v. 2.0.6; [70]), selecting the best-fit substitution model based on the ModelFinder [71] inference, allowing for polytomies and evaluating branch supports using 1000 ultrafast bootstrap (UFBoot; [72]) replicates.

The resulting tree was midpoint rooted and visualized in FigTree (Rambaut, A. FigTree v. 1.4.4; available online: http://tree.bio.ed.ac.uk/software/figtree/ accessed on 10 May 2021). Inkscape (v. 1.0.1; available online: https://inkscape.org accessed on 10 May 2021) was then used to combine, as well as annotate, the trees based on the phage complete-genome-associated metadata (host genus, taxonomical placement).

#### 3.7.4. Pairwise Genome/Proteome Comparisons with Closest Relatives

To investigate and visualize which regions of the genome/which proteins were similar and which differed greatly in comparison to the closest identified relatives of the studied *Pantoea* phages Nifs112, Nufs112, and Nafs113, Easyfig (nucleotide sequence comparison; v.2.2.2; [75]) and Clinker (protein sequence comparison; v.0.0.23; [76]) were selected for the task.

That way, the complete genome of the *Pantoea* phage Nifs112 (OK570184) was compared with two representatives of the genus *Eracentumvirus*—*Erwinia* phages vB_EamP-S2 (NC_047917) and Era103 (NC_009014).

The *Pantoea* phage Nufs112 (OK570185) was compared with the genomes of the *Klebsiella* phage 6939 (OL362271, rearranged to begin with base 6000) and the *Ralstonia* phage Reminis (MN478376, rearranged to begin with base 39,417 and reverse-complemented), both of which had to be rearranged taking the identified SDTR location in the genome of Nufs112 into account. Additionally, genus *Gajwadongvirus* representatives *Escherichia* phage ECBP5 (KJ749827) and *Pectobacterium* phage PP99 (NC_047802) were also used for Nufs112-related comparisons.

The *Pantoea* phage Nafs113 was compared with the sole representative of the genus *Lietduovirus, Pantoea* phage vB_PagM_LIET2 (NC_048751).

Generated figures were further visually refined in Inkscape (v. 1.0.1; available online: https://inkscape.org accessed on 10 May 2021).

For pangenome/proteome comparisons performed within this study, Roary (v. 3.13.0; [29]) was employed under the different BLASTP identity thresholds using the publicly available corresponding phage GenBank files that were transformed into the *.gff format with the help of the Bio:Perl script bp_genbank2gff3.pl [77].

## 4. Conclusions

Three bacteriophages infecting the same bacterial strain were isolated from an unidentified Latvian grasshopper species alongside its host *Pantoea agglomerans* LS5-2. The isolated phages were very different from each other and, to our knowledge, were so far the first three *Pantoea* sp.-infecting phages isolated from an insect source.

The *Pantoea* podophage Nifs112 (OK570184) was proposed to represent a novel species within the phage family *Autographiviridae*, subfamily *Molineuxvirinae*, genus *Eracentumvirus* (comprising so far exclusively phages infecting *Erwinia*).

The *Pantoea* podophage Nufs112 (OK570185) was evolutionary moderately related to two of the phages yet without a standing in the official phage taxonomy (*Ralstonia* phage Reminis, *Klebsiella* phage 6939) and, to a lower extent, to representatives of the phage genus *Gajwadongvirus*. The sufficient amount of difference Nufs112 showed in comparison to either of the so-far-sequenced phages, however, allowed us to propose it as an exemplar isolate for the creation of a new phage species that would represent a novel phage genus.

The *Pantoea* myophage Nafs113 (OK570186) had quite an unusual capsid with an approximately 3:1 length-to-width ratio. Nafs113 was very distantly related to any of the so-far completely sequenced phages that are publicly available, except for the *Pantoea* phage vB_PagM_LIET2, with whom Nafs113 shared a very close evolutionary relationship. For practicality reasons, at this time we propose to consider Nafs113 as an isolate of the phage species *Lietduovirus LIET2*, the so-far single phage species within the genus *Lietduovirus*, currently represented by the *Pantoea* phage vB_PagM_LIET2 alone.

Additional microbiological characterization of these three lytic phages will next be undertaken to determine their possible host ranges, lifecycle characteristics, and virion stability, ultimately to evaluate their potential to be used for biocontrol of not only *Pantoea* spp. but also of closely related bacteria (e.g., *Erwinia* spp.).

## Figures and Tables

**Figure 1 ijms-24-01820-f001:**
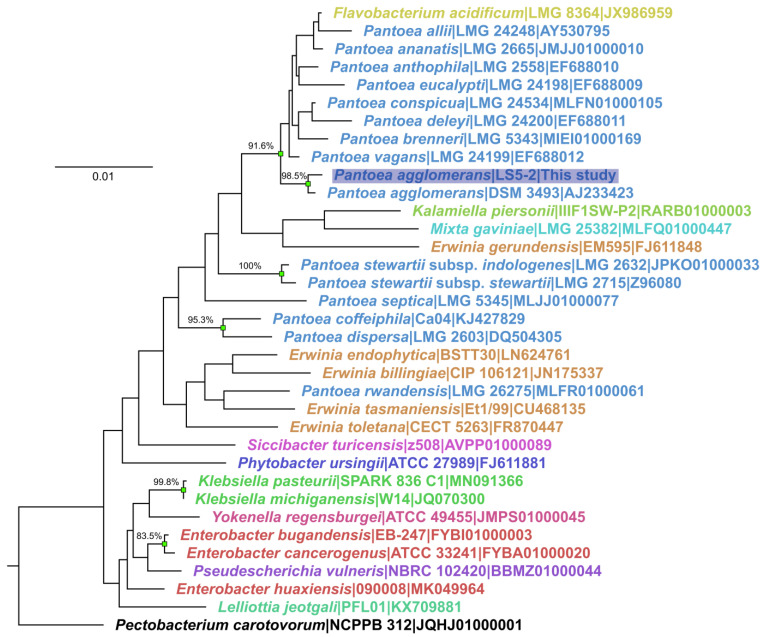
Neighbor-joining tree of 16S rRNA gene sequences of bacterial isolate LS5-2 and closely related bacterial species. The analysis involved 35 nucleotide sequences (16s rRNA gene sequences of *P. agglomerans* isolate LS5-2-host strain of the phages described within this study, 33 other closely related bacterial species, and *Pectobacterium carotovorum* as an outgroup for rooting). Tip labels correspond to the taxa and are in the format of “Species|Strain|Accession”; tip label colors correspond to different bacterial genera. The tip label of the sequence corresponding to the isolate LS5-2 is additionally highlighted in blue. The percentage of replicate trees in which the associated taxa clustered together in the bootstrap test (out of 1000 replicates) is indicated for branches having bootstrap support higher than or equal to 80%; such branches also have their distal nodes indicated by green rectangles. The evolutionary distances are in the number of base differences per site and the tree is drawn to scale. All positions with less than 90% site coverage were eliminated. That is, fewer than 10% alignment gaps, missing data, and ambiguous bases were allowed at any position. There were a total of 1346 positions in the final dataset.

**Figure 2 ijms-24-01820-f002:**
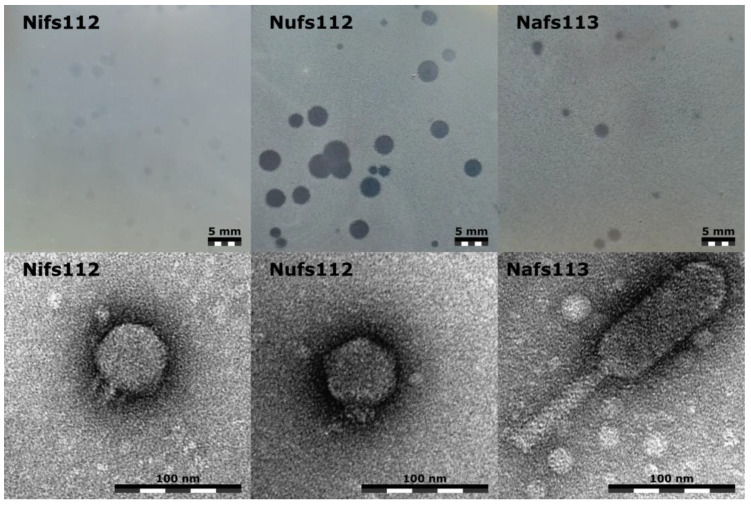
Plaque (**upper** row) and virion morphologies (**lower** row) of the studied *Pantoea* phages Nifs112, Nufs112, and Nafs113. Plaques were photographed after incubation for ~24 h at room temperature using *P. agglomerans* LS5-2 culture for the lawn and the same double agar overlay plating conditions; the scale bar represents 5 mm. Representative virion micrographs were obtained from the corresponding phage purified lysates negatively stained with 0.5% uranyl acetate; the scale bar represents 100 nm.

**Figure 3 ijms-24-01820-f003:**
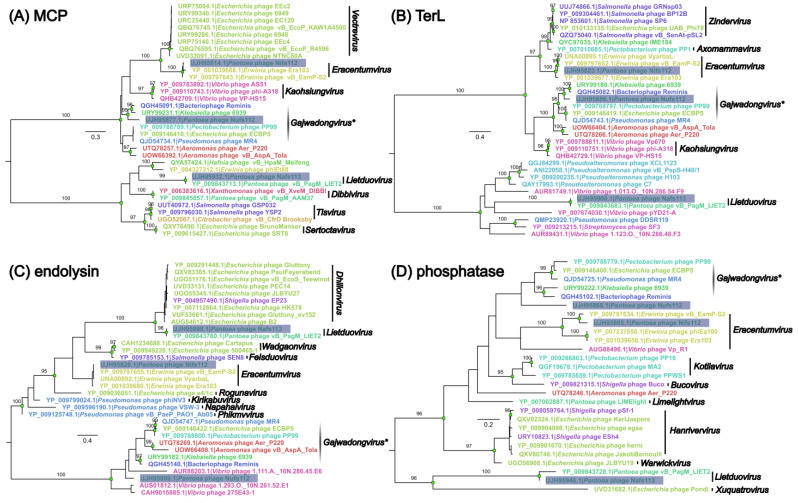
Midpoint-rooted maximum likelihood trees of the selected studied phage protein amino acid sequences and similar sequences found in the proteomes of other cultured phages. (**A**) Major capsid protein (MCP), (**B**) terminase large subunit (TerL), (**C**) endolysin, (**D**) phosphatase dataset; multiple sequence alignment and tree details can be found in Appendix A. The trees are drawn to their respective scales, and branch lengths correspond to the number of amino acid substitutions per site. Branches with UFBoot support of ≥95% (out of 1000 replicates) have their distal nodes indicated as green rectangles, and the corresponding UFBoot value is indicated above the branches. In all of the trees, tips are labeled as “Respective protein accession|Originating phage”, and tip labels are colored based on the respective phage host genus; labels of the leaves containing the amino acid sequences of the respective studied phage proteins are highlighted in blue. Labeled black bars after the tip labels show clade genus level annotations retrieved from the respective phage complete-genome metadata at the time of writing (if indicated). Cloudy black bar labeled “*Gajwadongvirus**” indicates dubious genus-level designation for some of the phages and contains some phages without genus-level associated taxonomy information.

**Figure 4 ijms-24-01820-f004:**
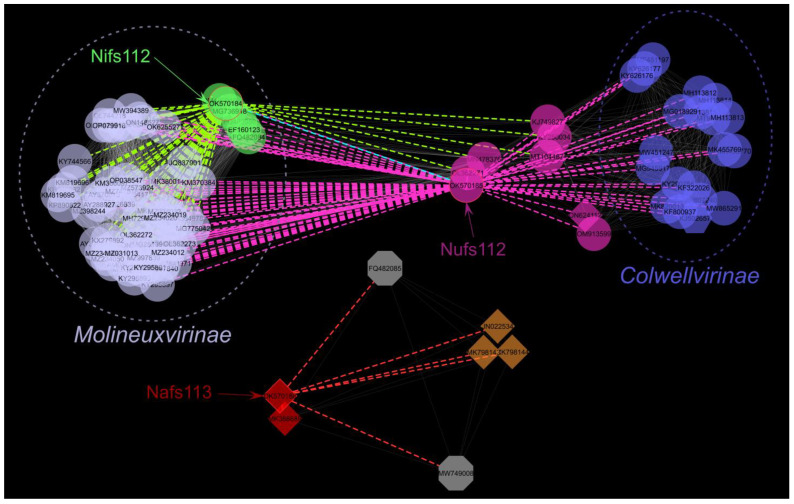
Sub-network containing first neighbors of the studied phages under edge-weighted spring-embedded layout (n = 103). Nodes are colored according to the viral cluster as identified by vConTACT2 for a network generated from 18,553 complete phage genomes: VC_436_0 (n = 62)—light blue; VC_437_0 (n = 6)—green; VC_961_0 (n = 20)—dark blue; VC_62_0 (n = 3)—brown; VC_794_0 (n = 2)—red; outliers (n = 2)—grey; phages that were determined to be classified as an overlap between two viral clusters (VC_439/VC_962; n = 8) that were not present in the sub-network—magenta. Different node shapes correspond to different viral families: circle—*Autographiviridae*; octagon—*Myoviridae* (recently abolished); hexagon—*Drexlerviridae*; diamond—unclassified at the family level. Dashed ellipses enclose recognized and tentative *Molineuxvirinae* (light blue) or *Colwellvirinae* (dark blue) phage subfamily representatives. Dashed colored edges represent links to the studied phages: green—Nifs112; magenta—Nufs112; red—Nafs113. Mutual link between Nifs112 and Nufs112 is represented by cyan dashed line.

**Figure 5 ijms-24-01820-f005:**
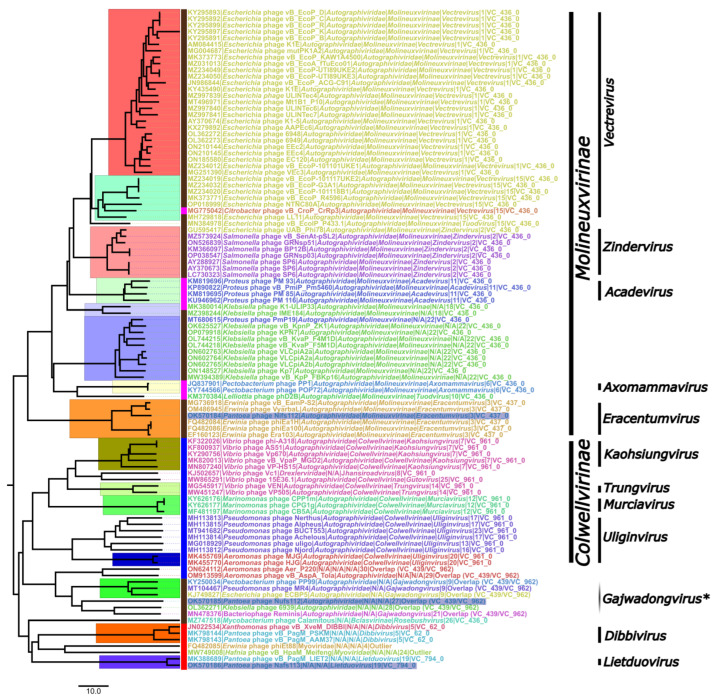
The NJ tree from a pairwise intergenomic distance matrix generated by VIRIDIC for a dataset comprising the studied phages and their first neighbors identified by the vConTACT2 analysis (n = 103). The tree is midpoint rooted and drawn to scale; the scale bar represents 10% genome nucleotide sequence divergence. Tip labels follow the format of “Genome accession|Phage|Family|Subfamily|Genus|VIRIDIC genus cluster (at least 70% intergenomic similarity)|vConTACT2 status or cluster”. Tip labels of the studied phages are highlighted in blue. The taxonomical information was retrieved from the genome accession-associated metadata at the time of writing. Black bars after the tips show phage subfamily- and genus-level annotations, where two or more corresponding rank phages were neighbors in the tree. Monophyletic clades corresponding to VIRIDIC genus-level clusters are also highlighted by arbitrarily colored rectangles in the tree. The color of the rectangle between the external leaf of the tree and the corresponding tip indicates the “first neighbor” status regarding the studied phages based on the vConTACT2 analysis and is as follows: brown—Nifs112 and Nufs112; magenta—Nifs112; blue—Nufs112; red—Nafs113.

**Figure 6 ijms-24-01820-f006:**
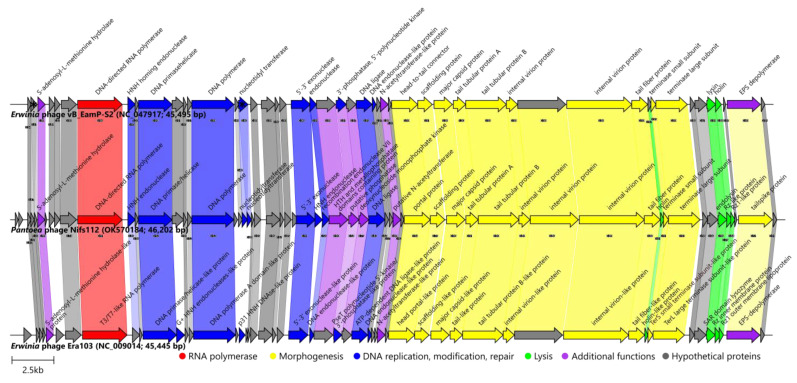
Genome organization and proteome content comparison of *Pantoea* phage Nifs112 to its closest relatives. Genomes are drawn to scale; the scale bar indicates 2500 base pairs. Arrows representing open reading frames point in the direction of the transcription and are color-coded based on the function of their putative product according to the legend. Slanted labels above the arrows indicate the predicted function for the given ORF putative product in the case where it had a function assigned (original annotations from downloaded GenBank files were retained). Ribbons connect phage proteins sharing >30% amino acid sequence similarity and are colored in lighter shades according to the predicted functional group of the respective *Pantoea* phage Nifs112 ORF product. Ribbon labels indicate the similarity between the connected ORF products and ribbons have their fill color opacity set to the percentage similarity between the respective proteins. Asterisks within two of the arrows representing vB_EamP-S2 ORFs indicate that their homologs in Nifs112 were encoded by two distinct ORFs in each case.

**Figure 7 ijms-24-01820-f007:**
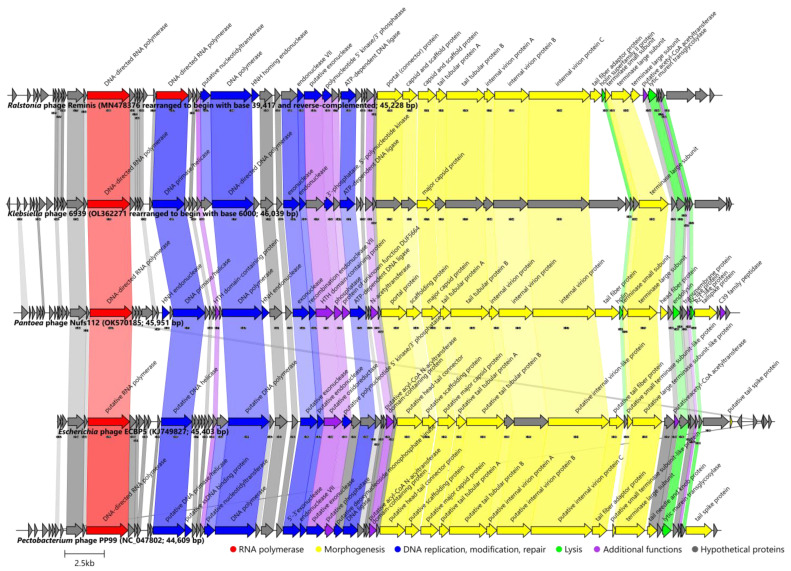
Genome organization and proteome content comparison of *Pantoea* phage Nufs112 to its closest relatives. Genomes are drawn to scale; the scale bar indicates 2500 base pairs. Genomes of *Ralstonia* phage Reminis and *Klebsiella* phage 6939 were rearranged to ensure collinearity with Nufs112, for which exact genome termini were identified. Arrows representing open reading frames point in the direction of the transcription and are color-coded based on the function of their putative product according to the legend. Slanted labels above the arrows indicate the predicted function for the given ORF putative product in the case it had a function assigned (original annotations from downloaded GenBank files were retained). Ribbons connect phage proteins sharing >30% amino acid sequence similarity and are colored in lighter shades according to the predicted functional group of the respective *Pantoea* phage Nufs112 ORF product. Ribbon labels indicate the similarity between the connected ORF products and ribbons have their fill color opacity set to the percentage similarity between the respective proteins.

**Figure 8 ijms-24-01820-f008:**
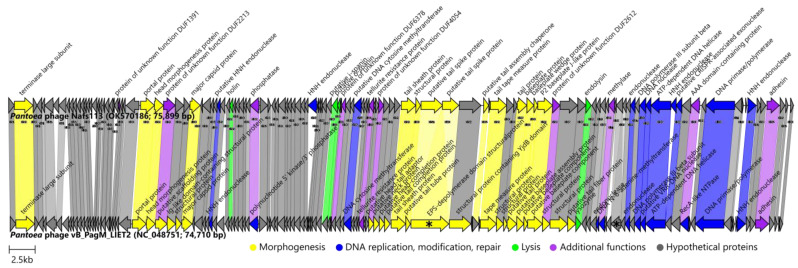
Genome organization and proteome content comparison of *Pantoea* phage Nafs113 to its closest relative—*Pantoea* phage vB_PagM_LIET2. Genomes are drawn to scale; the scale bar indicates 2500 base pairs. Arrows representing open reading frames point in the direction of the transcription and are color-coded based on the function of their putative product according to the legend. Slanted labels above the arrows indicate the predicted function for the given ORF putative product in the case it had a function assigned (original annotations from downloaded GenBank files were retained). Ribbons connect phage proteins sharing >30% amino acid sequence similarity and are colored in lighter shades according to the predicted functional group of the respective *Pantoea* phage Nafs113 ORF product. Ribbon labels indicate the similarity between the connected ORF products and ribbons have their fill color opacity set to the percentage similarity between the respective proteins. Asterisks within two of the arrows representing vB_PagM_LIET2 ORFs indicate that their homologs in Nafs113 are encoded by two distinct ORFs in each case.

**Table 1 ijms-24-01820-t001:** Overview of the so-far publicly available completely sequenced phages infecting bacteria from genus *Pantoea*. Rows containing information about phages Nifs112, Nufs112, and Nafs113 described in this study are in bold. “*Pantoea* phage” column provides a reference with further details about a particular phage, if such a reference could be found. Taxonomy was retrieved from the genome accession metadata, and the exclamation mark (!) after some of the family names indicates that this phage family was recently abolished by the ICTV. An asterisk (*) in the genus column indicates that the corresponding phage is not a part of the official phage taxonomy as per (VMR VMR_20-190822_MSL37.3). Isolation source column cells are colored based on the isolation source category: brown—soils; green—plant-associated material; yellow—insect-associated material; blue—water-associated source.

Genome Accession	*Pantoea* Phage (Reference)	Genome Length (bp)	GC%	ORFs	Family	Genus	Host	Isolation Source
FR687252.1	LIMElight [19]	44,546	53.991	55	*Autographiviridae*	*Limelightvirus*	*P. agglomerans*	soil from potato trial field
FR751545.1	LIMEzero [19]	43,032	55.352	57	*Autographiviridae*	*Waewaevirus*	*P. agglomerans*	soil from potato trial field
MG948468.1	vB_PagS_Vid5 [20]	61,437	48.821	99	not defined	*Vidquintavirus*	*P. agglomerans*	*Amelanchier spicata*
MK095605.1	vB_PagS_MED16 [21,22]	46,103	55.111	73	*Siphoviridae* (!)	not defined *	*P. agglomerans*	*Amelanchier spicata*
MK095606.1	vB_PagS_AAS23 [22,23]	51,170	47.645	92	*Drexlerviridae*	*Sauletekiovirus*	*P. agglomerans*	*Ribes* Jostabeere
MK388689.1	vB_PagM_LIET2 [22]	74,710	53.989	131	not defined	*Lietduovirus*	*P. agglomerans BSL*	*Amelanchier spicata*
MK770119.1	vB_PagS_AAS21 [22]	116,649	39.007	213	*Demerecviridae*	not defined *	*P. agglomerans* AUR	*Ribes* Jostabeere
MK798142.1	vB_PagM_AAM22 [24]	49,744	48.436	96	*Myoviridae* (!)	not defined *	*P. agglomerans* AUR	*Ribes* Jostabeere
MK798143.1	vB_PagM_AAM37	49,990	52.048	86	not defined	*Dibbivirus*	*P. agglomerans* AUR	*Ribes* Jostabeere
MK798144.1	vB_PagM_PSKM	49,935	52.356	82	not defined	*Dibbivirus*	*P. agglomerans* ARC	*Amelanchier spicata*
MN450150.1	vB_PagP-SK1 [25]	39,938	52.324	42	*Autographiviridae*	*Elunavirus **	*P. agglomerans* SN01121	barnyard soil
MT230534.1	vB_PagM_SSEM1	54,982	44.182	97	*Chaseviridae*	*Loessnervirus*	*P. agglomerans* strain SER	red currant
MZ501269.1	AH01	46,062	52.232	69	not defined	not defined *	*Pantoea* sp.	leaf of horse chestnut tree
MZ501270.1	AH07 [26]	37,859	52.220	58	*Myoviridae*(!)	not defined *	*Pantoea* sp.	leaf of horse chestnut tree
**OK570184.1**	**Nifs112** **[This study]**	**46,202**	**50.214**	**59**	* **Autographiviridae** *	* **Eracentumvirus *** *	* **P. agglomerans** *	**insect gut**
**OK570185.1**	**Nufs112** **[This study]**	**45,951**	**47.701**	**67**	* **Autographiviridae** *	**not defined** ** *** **	* **P. agglomerans** *	**insect gut**
**OK570186.1**	**Nafs113** **[This study]**	**75,899**	**54.067**	**130**	**not defined**	** *Lietduovirus ** **	* **P. agglomerans** *	**insect gut**
OL396571.1	PdC23	44,715	49.661	75	*Siphoviridae*(!)	not defined *	*P. dispersa* LMG2603	waste water
OL744209.1	vB_PdeP_F1M1C	38,645	50.734	55	*Autographiviridae*	*Teetrevirus **	*P. deleyi*	stormwater streams
OL744212.1	vB_PdeP_F2M1C	39,424	50.700	57	*Autographiviridae*	*Teetrevirus **	*P. deleyi*	stormwater streams
OL744217.1	vB_PdeP_F5M1C	36,790	50.155	56	*Autographiviridae*	*Teetrevirus **	*P. deleyi*	drain water
OL744220.1	vB_PdiM_F5M2A	149,913	50.579	320	not defined	*Certrevirus **	*P. dispersa*	drain water

**Table 2 ijms-24-01820-t002:** Overview of the *Pantoea* phage Nifs112, Nufs112, and Nafs113 genomes.

Phage	Virion Morphology	Genome Accession	Genome Length	Genome GC Content (%)	Genome Termini	Mean Genome Depth	Whole-Genome Coverage	ORFs	Hypothetical Proteins
Nifs112	Podophage	OK570184.1	46,202	50.2%	296 bp SDTR	223×	≥106×	59	28/59
Nufs112	Podophage	OK570185.1	45,951	47.7%	410 bp SDTR	685×	≥324×	67	35/67
Nafs113	Myophage	OK570186.1	75,899	54.1%	Circularly permuted	286×	≥30×	130	91/130

**Table 3 ijms-24-01820-t003:** Custom primers designed to verify the physical phage genome molecule termini of *Pantoea* phages Nifs112 and Nufs112.

Phage (Accession)	Primer	Primer Sequence (5′–3′)	Coordinates
Nifs112 (OK570184)	*ph112-1_Fw*	TGATGTATGCGTGTGTCAGC	45,884–45,903
	*ph112-1_Rv*	AGCCACTGTGTAGCCTAGG	314–332
Nufs112 (OK570185)	*ph112-2_Fw*	GGTCAGGAACTTACGTGAGG	45,521–45,540
	*ph112-2_Rv*	CCACCAGAGGTTGAGTGAC	422–440

## Data Availability

The annotated complete-genome sequence of the *Pantoea* phages Nifs112, Nufs112, and Nafs113 reported herein are available at GenBank under accession numbers OK570184.1, OK570185.1, and OK570186.1, respectively. The accession numbers of the other phage genomes and/or their product amino acid sequences used in the study are listed in either the respective figures/tables or Appendix A.

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
