# Peer review of "Three Phages One Host: Isolation and Characterization of Pantoea agglomerans Phages from a Grasshopper Specimen"

_ijms, 2023, doi:10.3390/ijms24031820_

Round 1

Reviewer 1 Report

Pantoea agglomerans is a ubiquitous Enterobacteriaceae that is found in plants and in the feces of humans and animals. The bacterium is also an opportunistic pathogen to some plants and animals. P. agglomerans has a wide range of adaptability and strong stress resistance, resulting in inconvenience to the prevention and control of its diseases. Bacteriophages have been regarded as biocontrol agents in medical treatment and agriculture. In this study, the authors isolated three bacteriophages from an unidentified Latvian grasshopper species, by using P. agglomerans, isolated from the same insect, as the target host. Transmission electron microscopy has revealed that one of the phages is a myophage with an unusual morphology, while the two others are typical podophages. The genome sequencing and comparative analysis indicated that the isolated phages are very different from each other and are the so far first three Pantoea sp. infecting phages isolated from an insect source. This study further expanded our knowledge about the bacteriophages and is helpful for developing new strategies and methods to control Pantoea infecting diseases.

Suggestions:

1.        Did the authors compare the phage genomes with the bacterial genomes in NR database at NCBI by BlastN? Have they noticed any homolous lysogenic phages in any bacterial genomes?

2.        In Figure 1, Bdellovibrio bacteriovorus as an outgroup for rooting is a little bit far in kinship with Pantoea agglomerans, which manifested an unnecessary long neck and crowded out the phylogenetic information of Pantoea clade.

Author Response

Comments and Suggestions for Authors

Pantoea agglomerans is a ubiquitous Enterobacteriaceae that is found in plants and in the feces of humans and animals. The bacterium is also an opportunistic pathogen to some plants and animals. P. agglomerans has a wide range of adaptability and strong stress resistance, resulting in inconvenience to the prevention and control of its diseases. Bacteriophages have been regarded as biocontrol agents in medical treatment and agriculture. In this study, the authors isolated three bacteriophages from an unidentified Latvian grasshopper species, by using P. agglomerans, isolated from the same insect, as the target host. Transmission electron microscopy has revealed that one of the phages is a myophage with an unusual morphology, while the two others are typical podophages. The genome sequencing and comparative analysis indicated that the isolated phages are very different from each other and are the so far first three Pantoea sp. infecting phages isolated from an insect source. This study further expanded our knowledge about the bacteriophages and is helpful for developing new strategies and methods to control Pantoea infecting diseases.

Author reply:

We thank the respected Reviewer 1 for reading through the manuscript and making suggestions aimed at further improving it.

  1. Did the authors compare the phage genomes with the bacterial genomes in NR database at NCBI by BlastN? Have they noticed any homolous lysogenic phages in any bacterial genomes?

Thank you for this question. We, indeed, did compare the studied phage genome nucleotide sequences with bacterial genomes present in GenBank (taxid: 2) using megablast, and all of them had a few hits, although nothing noteworthy, as query coverages for such hits for either of the three phages were less than a few percents of the phage genomes. Longest local alignments to bacterial genomes were <500 nt for Nifs112 (~68% identities); <700 nt for Nufs112 (~67% identities), <50 nt for Nafs113 (95% identities). Thus, they were considered to likely represent some conserved protein domain encoding nucleotide stretches that were not of interest within this study and were not inspected further. Although unsurprising, as all three of the studied phages are strictly lytic, we have opted to outline this lack of similarity with bacterial genomes in the main text as a follow-up to your question that could also have arisen from the other reader perspective.

Lines 555-559 of the revised manuscript now read: “When the studied phage genomes were queried against the bacterial sequences found in GenBank (taxid: 2) using megablast, hits to bacterial genomes were limited to up to a few hundred nucleotides (less than a few percent query coverages) for each of the phages, indicating a lack of prophages or the remnants of thereof highly similar to the studied phages in the so far sequenced bacterial isolates.”.

  1. In Figure 1, Bdellovibrio bacteriovorus as an outgroup for rooting is a little bit far in kinship with Pantoea agglomerans, which manifested an unnecessary long neck and crowded out the phylogenetic information of Pantoea clade.

We thank Reviewer 1 for this suggestion. After some discussions, we agree that using 16S rRNA sequence from Bdellovibrio bacteriovorus to root this tree, although very safe, might not be a very good option, considering it represents a phylum different from other bacteria seen in the tree, and it, indeed, unnecessarily “squeezes” the part of the tree of interest (Pantoea clade). Thus, for the revised version of the manuscript a 16S rRNA sequence from the type strain of Pectobacterium carotovorum (NCPPB 312) – an outgroup representing another family (Pectobacteriaceae) within the order Enterobacterales, that is more closely related to the family Enterobacteriaceae and Erwiniaceae representatives seen in the tree, was chosen. We believe that this still ensures that the outgroup selected for rooting is a rather “safe” choice, while the phylogenetic information among the EzBiocloud database hits to 16S rRNA sequence of the host of the studied phages is now presented in a visually way more comprehensible manner. The corresponding figure and its caption, as well as the relevant main text section, have been updated accordingly. Should Reviewer 1 still suggest taking another outgroup to root the tree - we will do our best to comply in the next round of revisions.

Additionally, we have once again looked through the text to try fixing the grammar mistakes/spelling/writing errors to further improve the language of the manuscript to the best of our knowledge. Several such erroneous instances were spotted and corrected using the track changes.

Reviewer 2 Report

Author Nikita et al describes "Three Phages One Host: Isolation and Characterization of Pantoea agglomerans Phages from a Grasshopper specimen"

The Paper is well-reported and suitable for publication. 

Please remove the supplementary Figure and put it in the supplementary material.

Some of the written content in the figures are not visible in Figure 3, Figure 4, Figure 5, Figure 6, Figure 7, and Figure 8

Line 782: Why the name Pectobacterium is italic as well as underlined?

Author Response

Comments and Suggestions for Authors

Author Nikita et al describes "Three Phages One Host: Isolation and Characterization of Pantoea agglomerans Phages from a Grasshopper specimen"

The Paper is well-reported and suitable for publication. 

Author reply:

We would like to thank Reviewer 2 for the positive evaluation of our efforts and commentaries raising concerns about particular things in the submitted version of the manuscript.

Please remove the supplementary Figure and put it in the supplementary material.

Some of the Supplementary Figures were retained in the submitted version of the manuscript for the sake of improved readability to facilitate the peer-review process. We are sorry if this has caused confusion, but, as was initially intended at this stage, the Supplementary Figures that were present in the submitted version are now removed.

Some of the written content in the figures are not visible in Figure 3, Figure 4, Figure 5, Figure 6, Figure 7, and Figure 8

We thank the respected Reviewer 2 for bringing this to our attention. As IJMS is a digital online-only journal, we have opted to prepare high-resolution figures that would contain the maximum amount of information while taking up the least possible area/space in the manuscript. The finer details of the mentioned figures were meant to be viewed “zoomed in”, which the quality of the figures should perfectly allow. Although this probably can be adapted by changing the mentioned figure sizes and orientations, we feel it would make the manuscript less approachable to the potential readership.

Speaking of Figure 4, many overlapping network node labels in Figure 4, are, indeed, unreadable, but this is an intrinsic limitation of this method of graph presentation/visualization, where node labels are actually rather rudimentary in places where they overlap (finer overview of the phages seen in Figure 4 and their clustering is provided in Supplementary table S5).

                Upon closer inspection, we, however, have noticed that the quality of the figures has suffered greatly in the PDF version of the manuscript demonstrating a substantial image quality downgrade that has likely happened during the rendering from the high fidelity *.docx version. This is especially relevant to figures 6, 7, and 8, where protein similarity ribbon labels are so pixelated upon zooming in that they are, indeed, unreadable at this point in the PDF version. We hope that the journal team would help to render the provided images without the loss of image quality in further revision rounds and production. Additionally, in Figure 5 studied phage tip label highlights were made more narrow not to overlap with other labels in the revised version.

Honestly, we would like to retain the mentioned figures as they are (but with the initially intended high resolution that allows to zoom in to explore all the details, which is now lower than provided initially). However, should Reviewer 2 our editorial team find our reasoning to be inappropriate (the main point being that the manuscript materials were intended for an online format having zooming capabilities), we welcome specific suggestions that would help improving the mentioned figure written content without the loss of information presented.

Line 782: Why the name Pectobacterium is italic as well as underlined?

This was a formatting mistake that slipped through the revision of the draft version of the manuscript before submission. We thank Reviewer 2 for spotting this issue, which has now been corrected in the revised version.

Additionally, we have once again looked through the text to try fixing the grammar mistakes/spelling/writing errors to further improve the language of the manuscript to the best of our knowledge. Several such erroneous instances were spotted and corrected using the track changes.

Reviewer 3 Report

Dear authors, I reviewed this paper and have the following comments

MAJOR

-the paper is fairly long and some of the details given in the methodology section could have been provided in the supplementary documents as an annex.

-you need to clarify what the putative potential use of the 3 phages you discovered is. Science is fascinating, but there needs to be a practical aim. It is also not clear to me whether these phages may be used to control the infection produced by the Gram-negative bug in plants only or also in humans (immunodeficient patients).

-some paragraphs need to be revised as they do not make much sense (e.g. lines 630,  639-641).

-why was the grasshopper species not identifiable and how was the grasshopper sourced?

MINOR

-line 9 pls delete the dash between both and benefitting

Author Response

Comments and Suggestions for Authors

Dear authors, I reviewed this paper and have the following comments

Thank you for reviewing our manuscript. After discussing with the co-authors and taking the commentaries of the other two reviewers into account, we feel that most of the major comments provided hereby would fail to improve the manuscript and are a matter of subjective opinion/preference.
Please, find out point by point response below.

MAJOR

-the paper is fairly long and some of the details given in the methodology section could have been provided in the supplementary documents as an annex.

We agree that the paper is longer than usual, this, however, mainly has to do with a choice to present all three of the isolated phages that are very different from each other and were isolated from the same source in the same manuscript elaborating on their genomics. Although the three studied phages could have been proceeded to be described in the manuscripts phage by phage as short individual reports, that would be a case of irrelevant “salami slicing”.

Currently, the methodology section takes up to ~240 lines of the manuscript which is currently ~1200 lines long, which makes it merely 20% of the content of the manuscript. As per MDPI IJMS instructions for authors, we have tried to describe the methods “with sufficient detail to allow others to replicate and build on published results”. Where relevant, we’ve tried to give references either to commercial kit protocols or previously published methodology, mentioning only the modification, if any. We find no way of shortening the methods or removing parts of them to supplementary materials or annexes while still following the journal guidelines. Frankly, we do not see much reason to do so, as even at this stage, we believe, interested readers that find our methodology descriptions (~20% of the content line-wise) far too long might just skip the methodology section as a whole and read further, returning to consult it only in case of interest. However, we do not oppose moving the whole material and method section to make it the last section of the manuscript, either before or after conclusions or even as an annex/supplementary document, if the editorial team would allow this.

That being said, we do not see shortening the section by moving its parts somewhere else or omitting them as an option, but we are not against moving the whole methodology section to the other part of the manuscript if the special issue editors/journal editors give us an allowance to do so.

-you need to clarify what the putative potential use of the 3 phages you discovered is. Science is fascinating, but there needs to be a practical aim. It is also not clear to me whether these phages may be used to control the infection produced by the Gram-negative bug in plants only or also in humans (immunodeficient patients).

“Science is fascinating, but there needs to be a practical aim” is a very blunt and debatable claim we, being especially fond of fundamental phage research, find outright offensive.

You know, honestly, it is not clear at this point to us (the people that worked with these phages for some time) either, whether these three phages may be used to control any infection at all. This is why the closing sentence of the conclusions reads “Additional microbiological characterization of these three lytic phages will next be partaken to determine their possible host ranges, lifecycle characteristics, and virion stability, ultimately to evaluate their potential to be used for biocontrol of not only Pantoea spp., but also of closely related bacteria (e.g. Erwinia spp.).” Evidently, at this point, we can clarify nothing about their potential use, as lots and lots of experimental work is still ahead to determine whether it is worth even trying to bring these three phages further than in vitro.

The answer to your question requires up to several years of extensive research on these phages specifically in aspects relevant to biocontrol. The aim of this paper, however, was to describe the isolation and mainly genomic characterization of the novel phages for a host that does not have many phages known to infect it yet. Although we do have some follow-up data regarding their virion stabilities, killing curves, etc., this is out of the scope of the current manuscript (which was perceived by you as “fairly long” already) and means nothing regarding the practical applicability of these phages as biocontrol agents before we can get our hands on a collection of different pathogenic Pantoea spp. and related bacterial strains and perform host range screens to determine how broad or narrow their host ranges are.

-some paragraphs need to be revised as they do not make much sense (e.g. lines 630,  639-641).

After reading through the mentioned lines of the submitted manuscript, we wholeheartedly disagree.

About this particular paragraph that “does not make much sense”… Having an alleged Drexlerviridae representative among part of the intergenomic distance NJ tree consisting of Autographiviridae representatives would not make sense without discussion lines explaining why this has occurred and highlighting the fact that one should not blindly trust biologically databases/repositories, thus this mentioned paragraph was written.

As crazy as the origin of this mistake is, we are adamant that it is ought to at least be mentioned, to make other researchers aware that such things are prone to happening as well. In the revised version, this paragraph was shortened greatly (currently, it takes 8 lines and 93 words, whereas in the original submission it was 299 words long and took 23 lines).

If there are other paragraphs perceived by the respected Reviewer 3 as “not making much sense”, please, let us know, and we will provide our rationale for their presence in the next round of revisions.

-why was the grasshopper species not identifiable and how was the grasshopper sourced?

None of the paper co-authors is an entomologist to be able to identify grasshoppers as well as we can identify phages :)

Jokes aside, a dead grasshopper was collected in a private field near Pociems, Latvia, and was viewed merely as an intriguing source to look for novel phages, thus, we were not even remotely interested in its exact taxonomical placement. We would say that any insects are still a rather undersampled cultured phage source, which is in line with our recent reports of interesting novel phages originating from insects such as moths (e.g. https://doi.org/10.3390/microorganisms9071540), bees (e.g. https://doi.org/10.3390/microorganisms10091799; https://doi.org/10.1007/s00705-019-04516-2), as well as mixed insects (e.g. https://doi.org/10.3389/fmicb.2020.01245).

MINOR

-line 9 pls delete the dash between both and benefitting

Thank you, deleted it.

Additionally, we have once again looked through the text to try fixing the grammar mistakes/spelling/writing errors to further improve the language of the manuscript to the best of our knowledge. Several such erroneous instances were spotted and corrected using the track changes.